

# Observing ice particle growth along fall streaks in mixed-phase clouds using spectral polarimetric radar data

Lukas Pfitzenmaier[1], Christine M. H. Unal[1], Yann Dufournet[2], and Herman J. W. Russchenberg[1]

[1]TU Delft, Civil Engineering and Geosciences, Geosciences and Remote Sensing, Stevinweg 1, 2628 CN Delft, The Netherlands
[2]SkyEcho, HD weather nowcasting, Delft, The Netherlands

*Correspondence to:* Lukas Pfitzenmaier (l.pfitzenmaier@uni-koeln.de)

**Abstract.** The growth of ice crystals in presence of super-cooled liquid droplets represents the most important process for precipitation formation in the mid-latitudes. Such mixed-phase interaction processes remain however pretty much unknown, as capturing the complexity in cloud dynamics and microphysical variabilities turns to be a real observational challenge. Ground-based radar systems equipped with fully polarimetric and Doppler capabilities in high temporal and spatial resolutions

such as the S-band Transportable Atmospheric Radar (TARA) are best suited to observe mixed-phase growth processes. In this paper, measurements are taken with the TARA radar during the ACCEPT campaign (Analysis of the Composition of Clouds with Extended Polarization Techniques). Besides the common radar observables, the 3D wind field is also retrieved due to TARA unique three beam configuration. The novelty of this paper is to combine all these observations with a particle evolution detection algorithm based on a new fall streak retrieval technique in order to study ice particle growth within complex

precipitating mixed-phased cloud systems. In the presented cases, three different growth processes of ice crystals, plate-like crystals, and needles, are detected and related to the presence of supercooled liquid water. Moreover, TARA observed signatures are assessed with co-located measurements obtained from a cloud radar and radiosondes. This paper shows that it is possible to observe ice particle growth processes within complex systems taking advantage of adequate technology and state of the art retrieval algorithms. A significant improvement is made towards a conclusive interpretation of ice particle growth processes

and their contribution to rain production using fall streak rearranged radar data.

## 1 Introduction

The interaction of liquid water droplets and ice crystals leads to an enhanced growth of the ice crystals while falling through layers of super-cooled liquid water. In the mid-latitudes, this is an important process for precipitation formation (Pruppacher and Klett, 1996; Lamb and Verlinde, 2011). Nevertheless, the implications of the microphysical processes on precipitation for-

mation is currently not well understood. Such microphysical processes involve a variety of hydrometeor sizes, shapes, phases, all affected by cloud dynamics, making the observation particularly challenging (Kollias et al., 2007; Shupe et al., 2008). Nowadays, ground-based radar measurement techniques have the advanced capabilities to observe and study microphysical processes within mixed-phase cloud systems (Kollias et al., 2007). Analyses of ice particle growth processes within mixed-phase cloud systems are primarily based on cloud radar data. Among them, Oue et al. (2016) uses polarimetric radar ($K_a$-, and





X-band) and lidar measurements to study ice particle growth processes in the cloud. Kalesse et al. (2016) analyzes Doppler spectra ($K_a$-band) during a riming event and compare their findings of particle growth rates from the observations to simulations of such a riming even. Data from the same measurement campaign are used by (Kneifel et al., 2015), where they apply a radar triple frequency method (W-, $K_a$-, and X-band) to distinguish between different ice hydrometeor types (aggregates, rimed particles) and their particle size distribution characteristics during snowfall events. Myagkov et al. (2016) uses polarimetric radar measurements ($K_a$-band) to derive the shape and orientation of mainly pristine ice particles generated at liquid topped mixed phase clouds.

In the studies mentioned above, the focus is on stratiform and layered clouds (dynamically stable and homogeneous) in the Arctic region or on snowfall events that do not involve any phase transition (ice particles constantly remain in the ice phase). Studies of precipitating mixed-phase cloud systems where melting of the ice particles is involved are rare. However, to understand the whole precipitation formation process the growth of the ice particles and the associated microphysical processes above the melting layer have to be studied to be related to the precipitation intensity below the melting layer. Nevertheless, the frequency bands listed in the studies above (W-, $K_a$-, and X-band) suffer from attenuation when it comes to observations of precipitation cloud systems, Kollias et al. (2007). To avoid attenuation in precipitation, radars that operate in lower radar frequencies (C- and S-band) can be used. In the past, low-frequency radars were mainly used to estimate precipitation (Bringi and Chandrasekar, 2001) and to classify hydrometeor types (Straka et al., 2000; Chandrasekar et al., 2013). However, in the last few years, the polarimetric capabilities of those radars were also used to study ice particle microphysics and growth processes in cloud systems (during snowfall events or in the ice part of the cloud above the melting), Kennedy and Rutledge (2011); Bechini et al. (2013); Moisseev et al. (2015). Microphysical studies of ice crystal habits using a C-band radar were first done by Bader et al. (1987). They compared differential reflectivity, $Z_{DR}$, signatures from the radar with aircraft measurements of ice crystal habits of precipitating stratiform clouds. It was found that the enhanced bands of $Z_{DR}$ values occur where the aircraft measurements detected defined pristine ice particle shapes. Later it was found that a combination of $Z_{DR}$ and specific differential propagation phase, $K_{DP}$, can be used to identify regions of plate-like particle growth processes and onset of aggregation (Kennedy and Rutledge, 2011; Moisseev et al., 2015). Andrić et al. (2013) modeled polarimetric signatures for a winter storm case. The related simulations showed agreements with the observations; nevertheless, the model was not able to match all the features of the observations in a single run.

Low-frequency radar observation of ice particle growth process due to the interaction of the ice crystals with supercooled liquid droplets is also found in the literature. Field et al. (2004) compared the observed $Z_{DR}$ signatures (C-band) to the obtained ice to liquid particle fraction measured with an aircraft and found a good agreement between liquid water presence and $Z_{DR}$ signatures that were caused by defined ice crystal shapes. Similar findings have also been made by Hogan et al. (2002). Hogan et al. (2002) showed that small convective cells embedded in warm frontal mixed-phase cloud systems triggered ice particle formation, growth, and multiplication related to super-cooled liquid water. Chandrasekar et al. (2013) discussed spectrograms measured with a C-band radar that exhibit signatures of aggregation and riming of ice particles. This shows that precipitation radar can capture particle growth in spectral domain and their related microphysics.





In this work, radar data measured during the ACCEPT campaign are presented. They are obtained with TARA, which is a fully polarimetric S-band Doppler radar profiler providing measurements in high temporal and spatial resolutions. Because of TARA's unique 3 beam configuration, the full 3D wind vector per sampling volume is also retrieved for each measurement. Therefore, using this radar and its configuration, it is possible to identify and study microphysical processes of ice particles

within complex mixed-phase clouds. On top of that, TARA measurements are rearranged along fall streaks according to the recently developed fall streak retrieval algorithm (Pfitzenmaier et al., 2017). Fall streaks are considered to be the precipitating path of a population of particles from top to bottom of the cloud system. Therefore, the analyzes of microphysical changes along the fall streaks contain information of microphysical evolution of the same particle population (Pfitzenmaier et al., 2017; Kalesse et al., 2016). The retrieval technique reconstructs the precipitation path based on the obtained TARA wind

information. In a next step, Doppler spectra and polarimetric radar variables rearranged along those fall streaks are used to study the microphysical evolution of a particle population. This paper demonstrates the advantage of analyzing data along fall streak instead of vertical profiles of radar data. Rearranged spectrograms of polarimetric variables allow to understand how the ice particle size and shape distribution changes versus height. This is used to improve the understanding of the impact of super-cooled liquid water on ice particle growth within a cloud system. Furthermore, the spectral signatures are

compared to additional measurements performed during the ACCEPT campaign ($K_a$-band cloud radar spectral information and radiosonde temperature profiles) to assess the results from TARA data. After introducing the observation strategy of the ACCEPT campaign, in Section 2, some examples of microphysical observations of ice particle growth processes and their spectral signatures are given in Sections 3 and 4. An overview of the fall streak technique is given in Section 5. Section 6 finally combines the retrieved microphysical information with fall streak correction to interpret growth processes of three

different cases. Discussions and conclusions are provided in Section 7.

## 2 Observation strategy

### 2.1 Instrumental setup of the ACCEPT campaign

The aim of the ACCEPT campaign (Analysis of the Composition of Clouds with Extended Polarization Techniques) is achieving a better understanding of microphysical processes involved in mixed-phase clouds using high-resolution polarimetric ob-

servations. The measurements were performed from October to mid-November 2014 at the Cabauw Experimental Site for Atmospheric Research (CESAR), the Netherlands. An instrumental synergy was used during ACCEPT to detect the different phases, the variety of sizes and shapes of the involved hydrometeors. For this purpose the TARA radar (Heijnen et al., 2000) at the site was extended by several other sensors: the Leipzig Aerosol and Cloud Remote Observation System (LACROS) (Bühl et al., 2013), a second vertically pointing $K_a$-band cloud radar Mira (Görsdorf et al., 2015), and the Raman Lidars Polly$^{XT}$

(Engelmann et al., 2016; Baars et al., 2016) and CAELI (CESAR Water Vapour, Aerosol and Cloud Lidar) (Apituley et al., 2009). A picture of the setup is provided in Figure 1. During special observation periods, radiosondes were launched at the site.





Some results of the ACCEPT campaign are already published. Myagkov et al. (2016) developed a retrieval to obtain the shapes of new generated ice crystals at liquid topped single layer mixed-phase clouds and compared their results to laboratory studies. Good agreement was obtained. Pfitzenmaier et al. (2017) developed a retrieval algorithm to reconstruct fall streaks within the TARA measurements to improve the study of ice particle growth due to supercooled liquid water presence within
the cloud system.

### 2.2   The Transportable Atmospheric RAdar - TARA

TARA is a frequency modulated continuous wave (FM-CW) S-band radar profiler ($3.3\,GHz$) that has full polarimetric and Doppler capabilities. TARA is able to provide high-resolution column measurements in the Doppler domain ($0.03$ m s$^{-1}$), in range ($30.0$ m, max. height of $10.05$ km) as well as high temporal resolution ($2.56$ s). Measurements are performed under a
fixed antenna elevation ($45°$) and a fixed azimuth ($246.5°$ related to the North). Measurements at $45°$ elevation are considered as an optimum to get polarimetric and Doppler spectra information related to microphysical properties of the probed medium (Moisseev et al., 2004; Unal et al., 2012). Therefore, TARA does not only allow the analysis of polarimetric bulk parameters as differential reflectivity, $Z_{DR}$, and co-polar correlation coefficient, $\rho_{HV}$, but it is also possible to study changes in the signatures of Doppler spectra, $sZ$, and spectral differential reflectivity ,$sZ_{DR}$ ($sZ_{DR}$ being the ratio of horizontal polarized to vertical
polarized spectral reflectivity, $sZ_{hh}/sZ_{vv}$). These changes in the spectra and spectrogram (height profiles of spectra) can be related to cloud microphysical variations. Therefore, it is possible to identify separate particle modes (having different velocity ranges) in the $sZ$ and $sZ_{DR}$ spectra. Furthermore, the unique three-beam configuration of TARA allows retrieving the full 3D wind vector per measurement. Using the Doppler spectra information of the three beams, main beam and two offset beams, the horizontal wind velocity $vd_h$, the vertical Doppler velocity $vd_V$, and the wind direction $\phi_W$ are retrieved with a minimal
temporal resolution of $2.56$ s (Unal et al., 2012). The combination of all these measurements makes it possible to study ice particle growth within such complex precipitating mixed-phase cloud systems. Table 1 gives an overview over the specifications of TARA during ACCEPT.

### 2.3   Instrumental synergy

In addition to the TARA radar, other co-located sensors measured in parallel, allowing full comparison of the observed data
signatures for similar volumes of study. In this paper, measurements of the Mira and TARA radars are compared, and data from two radiosonde launches are used. Unlike TARA, the data of high-frequency, Mira type, radar systems, suffer from attenuation during precipitation events. Low-frequency radar systems with high spatial and temporal resolution like the TARA radar have advantages measuring precipitation cloud systems. Nevertheless, the K$_a$-band radar is more sensitive to the smaller pristine ice particles near cloud top. Therefore, it can detect particles in far ranges where TARA, operating at a low-frequency, is not
able to measure these small particles. Despite of the attenuation the detected cloud top with Mira is at least $0.5$ km higher than the one identified with TARA. TARA and the vertical pointing Mira obtain linear depolarization ratio in the spectral domain ($sL_{DR}$) at high temporal resolution (1 s for Mira). This parameter, which is the ratio of $sZ_{vh}$, cross-polar received spectral reflectivity, to $sZ_{hh}$, is not affected by attenuation issues at vertical incidence. Therefore, comparisons between the



observed spectral polarimetric signatures from TARA and Mira can be done. These spectral polarimetric measurements are used in combination with thermodynamical conditions (temperature and humidity) within the cloud system, measured from radiosonde launches. On the one hand, radiosondes can retrieve regions of super-cooled liquid water within precipitating cloud systems when other methods or sensors are limited. On the other hand, the temperature profiles in super-cooled liquid layer and the polarimetric radar signatures can help identifying the dominant ice particle shape affecting the growth process in the mixed-phase area (Bader et al., 1987; Pruppacher and Klett, 1996; Fukuta and Takahashi, 1999; Myagkov et al., 2016).

## 3 Ice particle growth processes

Ice particles within clouds can grow through three main processes (Pruppacher and Klett, 1996; Lamb and Verlinde, 2011).

- *Riming of ice particles*: occurs when supercooled water droplets collide and freeze onto bigger ice crystals, see Figure 2 a). Ice particles grow into large, dense and almost spherical shaped particles. In terms of conical graupel the resulting particles get slightly prolate (Oue et al., 2015a). Because of the sensitivity of the reflectivity to the hydrometeor size, $Z_{hh} \sim \int D^6 N(D) dD$ where $N(D)$ is the hydrometeor size distribution and $D$ is the equivolumetric diameter, the growth of particles is strongly connected to an increase of reflectivity. The differential reflectivity $Z_{\mathrm{DR}}$ gives information on the shape of the measured particle population (positive = oblate, $\approx 0\,\mathrm{dB}$ spherical, and negative = prolate particle shape). The conceptual change in particle shapes is indicated by the dotted lines in Figure 2. Thus, rimming processes affect $Z_{\mathrm{DR}}$ values by turning horizontally aligned ice particle ($Z_{\mathrm{DR}} > 0$) into near spherical ($Z_{\mathrm{DR}}$ around 0) or slightly prolate particle ($Z_{\mathrm{DR}} < 0$) for conical graupels.

- *Water vapor diffusional growth of ice particles*: occurs when water vapor diffuses towards the crystals instead of forming supercooled droplets, see Figure 2 b). During that process particles grow, therefore the reflectivity increases. During the diffusional growth particles keep their characteristic shape, Pruppacher and Klett (1996); Fukuta and Takahashi (1999); Lamb and Verlinde (2011). Their $Z_{\mathrm{DR}}$ values slightly increase during the growth process. If particles grow large enough the crystals start to aggregate what again lead to a decrease of polarimetric signature.

- *Aggregation of ice particles*: occurs when ice crystals collide and form bigger ice crystals, see Figure 2 c). Aggregation leads to an increase of ice particle size and a change in particle shape from pristine particles to spherical shaped ice particles. The decrease of the differential reflectivity signatures depends strongly on the aggregated pristine particle type. Moisseev et al. (2015) reported that early aggregates of dendrites align horizontally and therefore can also contribute to a high $Z_{\mathrm{DR}}$-signature. This leads to a less strong decrease of polarimetric signatures of $Z_{\mathrm{DR}}$ with increasing size of the grown aggregates. Hobbs et al. (1974) reported that needles have the tendency to clump very fast into spherical particles if their number concentration is high enough. Therefore, needles that aggregate turn faster into spherical shaped particles than dendrites.



## 4 Signatures of ice particle growth in spectral radar observations

The main three ice particle growth processes, riming, diffusional growth and aggregation (Pruppacher and Klett, 1996; Lamb and Verlinde, 2011) result in different signatures in the spectral radar observations, see Figure 3. Identifying the growth process signatures in the Doppler spectrum, $sZ$, and differential reflectivity spectrum, $sZ_{DR}$, is challenging due to concurrent growth

process mechanisms occurring within the same resolution volume on different group of particles. Therefore, the signatures of different processes can be overlaid and difficult to separate.

Figure 3 a) sketches riming signatures in the Doppler spectrum, $sZ$ (black line), and differential reflectivity spectrum, $sZ_{DR}$ (red line). A Doppler spectrum represents the Doppler velocity distribution weighted by particle backscattering. Therefore, $sZ$ and $sZ_{DR}$ are related to the particle size distribution (negative velocities indicate particle movements towards the radar,

therefore, larger particles are associated with larger negative velocities), while $sZ_{DR}$ is related to the particle shape distribution. In a Doppler spectrum, a separated particle mode of larger and denser ice particles is often a clear indication of a riming process (Kollias et al., 2007; Chandrasekar et al., 2013; Oue et al., 2015a). As described in Figure 3 a) they form a separated particle mode in the Doppler spectrum (left side) that indicates larger particles fall velocities in comparison to already existing particle mode. The particles in the mode on the right side are growing. Ice crystals must have a certain size to be effective

to rime (Wang and Ji, 2000; Ávila et al., 2009). Due to the fast growth of already large ice crystals the reflectivity values are also quite large in the right riming mode. Because not all ice crystals fulfill this criteria and might grow due to diffusional growth and/or aggregation, in the left mode the reflectivity also increases. In comparison to $sZ_{DR}$ values measured in rain, where the oblateness of large rain drops produces the highest $sZ_{DR}$ values, well defined shapes of pristine crystals are mainly responsible of the $sZ_{DR}$-signatures in cloud. Riming of particles leads to a decrease of polarimetric signatures and a decrease

of the $sZ_{DR}$ values towards zero. In case that riming leads to conical graupel $sZ_{DR}$ values become negative (see dotted red line at the left particle mode). In the right mode, a clear shape-size dependence is still expected and the higher $sZ_{DR}$ values corrrespond to the smaller ice crystals that have kept their pristine ice crystal shape.

Figure 3 b) depicts a signature of diffusional growth in $sZ$ and $sZ_{DR}$. The small particle mode on the right side in the Doppler spectrum indicates the diffusional growth of the ice particles. The Doppler spectrum width increases due to the

aggregation mode of the larger particles. Particles that grow via water vapor diffusion keep their pristine crystal shape and, therefore, the $sZ_{DR}$-values of that mode stay large. The onset of aggregation is seen in the significant decrease of values in the $sZ_{DR}$ spectrum, because aggregation leads to more spherical shaped particles. A significant feature of the diffusional growth is the separated mode in the $sZ_{DR}$ spectrum. Such a mode of defined particle shapes is not present during the aggregation process of particles.

Figure 3 c) illustrates an aggregation signature of ice crystals in $sZ$ and $sZ_{DR}$. Due to the domination of the large particles in the signal the spectral reflectivity can be skewed towards bigger particles. Depending on the particle size distribution a separation of the different particles modes is not always possible in the Doppler spectrum. Therefore, the signatures in $sZ_{DR}$ are needed to separate the different particle modes in the spectral domain if they are present. Nevertheless, it has to be mentioned that the differential reflectivity is also influenced by the density of the measured particles. Therefore, the strong decrease of



$sZ_{DR}$ signature due to aggregation onset can be caused also by the lower density of the aggregates, compared to the pristine crystals.

The shown signatures in the spectral domain are used to identify qualitatively particle growth within mixed phase clouds. Using bulk and spectral polarimetric measurements, several cases during a complex precipitation system will be investigated
next.

## 5   Methodology

The aim of this study is to analyze the microphysical process of a particle population on its way from cloud top to the bottom of the cloud system. Therefore, radar data are rearranged along so-called fall streaks before their signatures are examined. This offers a new insight on ice growth processes occurring in complex, local and inhomogeneous cloud conditions in precipitating
cloud systems.

Marshall (1953) and Browne (1952) were the first that analyzed fall streak signatures within radar measurements. They investigated the structure and shape of outstanding reflectivity patterns within radar measurements. They described a fall streak like Yuter and Houze (2003), as a manifestation of an inhomogeneity in the microphysical structure of a cloud system. To be observed, the relative size and number of precipitation particles within the fall streak need to be large such that their radar
reflectivity stands out as a local maximum from the immediate background reflectivity. Nevertheless, it was already pointed out that the main shape and structure of the fall streaks is influenced by the present cloud dynamics (Marshall, 1953; Browne, 1952). Because of this dynamical influence on the fall streak shape and structure, Pfitzenmaier et al. (2017) defines a fall streak as the path of a particle population obtained from the observation of its own motion. Taking into account the dynamical conditions within the cloud system the fall streak retrieval is based on the mean 3D wind field retrieved by TARA. Therefore
the fall streak retrieval is based on radar Doppler measurements only.

Using directly measured 3D wind information (vertical Doppler velocity $vd_V$, horizontal wind speed $vd_h$, and wind direction $\phi_W$) the fall streak retrieval estimates the time displacement per height to reconstruct the path of a particle population. This is done using a bottom-up approach starting at the lowest (closest to the ground) valid data point at starting time $t_0$. The time displacement at the height $z$ is in principle estimated following Equation 1. It consists of two terms:

$$\Delta t(z) = t_0(z_0) + \sum_{z_0}^{z} \Delta t_\alpha(z_i) + \sum_{z_0}^{z} \Delta t_{dyn}(z_i) \tag{1}$$

The first term is the displacement time related to the antenna elevation, $\Delta t_\alpha$. The second term is the displacement time due to the cloud system dynamics, $\Delta t_{dyn}$. The calculation of the displacement contributions $\Delta t_\alpha$ and $\Delta t_{dyn}$ is done following the steps of Pfitzenmaier et al. (2017). The initial condition for the retrieval algorithm (cloud base height and averaging time for the wind profiles) are set individually per case, applying the suggested method in Pfitzenmaier et al. (2017).
The microphysical analyzes of the cases are based on fall streak corrected bulk parameter profiles of reflectivity, $Z$, differential reflectivity, $Z_{DR}$, and the co-polar correlation coefficient $\rho_{HV}$ as well as on the fall streak corrected Doppler spectra, $sZ$, and spectral differential reflectivity, $sZ_{DR}$. The growth of the measured particle population is indicated by an increase of





reflectivity. Changes in the $Z_{DR}$-signature give information on the shape and density changes of the measured particle popula-
tion. The co-polar correlation coefficient $\rho_{HV}$ gives information about particle homogeneity within the measured volume. The
closer $\rho_{HV}$ is to 1 the more homogeneous are the particle shapes of the measured particle population. Therefore, changes of
those parameters are used to examine the microphysical evolution of the particle population tracked from cloud top to bottom

(Pfitzenmaier et al., 2017; Kalesse et al., 2016; Oue et al., 2015a). The analysis of the fall streak corrected spectrograms are
used to investigate the changes in the spectra due to microphysical changes of the tracked ice particle distribution. Spectrograms
provide Doppler spectra per height bin at each time step (spectral reflectivity versus Doppler velocity and height). Therefore,
it is possible to identify the different signatures of riming, aggregation and diffusional growth of different pristine crystals by
analyzing their changes in signature (Figure 3) from cloud top to bottom.

Due to the variation of small time scale dynamics (horizontal and vertical wind at $45°$ elevation) on the spectra, the signatures
are only analyzed qualitatively. Nevertheless, the changes in the Doppler spectra shape (broadening, modality, and amplitude)
with height are detailed enough to provide information about the present particle microphysics. To ensure a good data quality
the spectra are averaged over 3 time bins per height (one time bin before and one time bin after the rearranged spectrum are
used for the averaging) as well as a $10$ dB threshold above the noise level is applied to Doppler bins in the spectra. This

removes artifacts at the edges of the spectra due to low SNR. However $sZ_{DR}$ values are still noisy. Therefore the presented and
analyzed visible signatures in the spectrograms are also checked manually for consistency over time and height.

## 6    Observations and results

Figures 4 and 5 show measurements obtained with the S-band radar profiler TARA, on $7^{th}$ November 2014, from 1000 UTC
to 1200 UTC.

In addition, the fields of reflectivity and linear depolarization ratio from the co-located vertical profiling cloud radar Mira
are presented in Figure 4 a) and d), respectively. The reflectivity ($Z$) fields in Figure 4 a) and b) display a precipitating cloud
system. The band of enhanced $Z$ values around $2.2$ km indicates the melting layer of this frontal system. The attenuation due
to precipitation is clearly visible for the $35$ GHz radar Mira, comparing the $Z$ fields of both radars (e.g. case 1 and 3 with
differences in $Z$ above the melting layer up to $20$ dB$Z$). The $Z$ fields in the cloud part show a high variation. Variability in

the structure of the fields of differential reflectivity ($Z_{DR}$) and linear depolarization ratio ($L_{DR}$) are also visible in Figure 4 c)
and d). They have a high correlation with the visible fall streak signatures in the $Z$-field (e.g. 1030 UTC - 1130 UTC above
$4$ km). Unfortunately, the TARA obtained wind fields between 1030 UTC and 1120 UTC are corrupted (due to clutter in the
non polarimetric offset beam measurements that influenced the retrieved wind fields especially the vertical Doppler velocity
component, see Figure 5 c)). Therefore, the data analysis is restricted to cases where the wind retrieval quality is good enough

to apply the fall streak retrieval technique (Pfitzenmaier et al., 2017). Figure 5 displays the Doppler spectrum width and wind
fields. Besides some small fluctuations in the horizontal wind field, Figure 5 b), the measurements show homogeneous wind
conditions within the cloud. In the wind direction field, Figure 5 d), a shear in wind direction is visible (about $30$ ° from
1000 UTC onwards). Later the wind direction shear reaches the cloud part (starting at 1135 UTC). Therefore, enhanced values



of vertical Doppler velocity and horizontal wind speed are visible as well as an increase of the Doppler spectral width can be identified. The turbulence is caused by the mixing of an approaching air-mass that is related to a cold frontal cloud system, into the present airmass. Case 1 analyses an example during the enhanced $Z_{\mathrm{DR}}$ and $L_{\mathrm{DR}}$ signatures on top of the lower cloud layer between 1009 UTC and 1018 UTC. While Case 2 focuses on the band of enhanced $Z_{\mathrm{DR}}$ between 1128 UTC and 1133 UTC

at around 3 km. For the last case, Case 3, the fall streak structures of $Z$ and polarimetric variables that generated within the enhanced band of $Z_{\mathrm{DR}}$ at around 5 km between 1135 UTC and 1147 UTC, are discussed.

### 6.1   Case 1: aggregation of needles, 1009-1018 UTC

With Case 1 the analysis of along a fall streak rearranged S-band radar data is discussed to understand the present particle growth process. Figure 6 a) shows the result of a retrieved fall streak at 101251 UTC. The results are obtained using a fixed

cloud base height of 2250 m and an averaging window of 30 s for the wind profile as initial conditions for the algorithm (Pfitzenmaier et al., 2017). The main features of interest are the enhanced $Z_{\mathrm{DR}}$ and $L_{\mathrm{DR}}$ signatures near the cloud top of the lower cloud layer (3.75 km to 3.5 km) between 1009 UTC and 1018 UTC in Figure 4. The reflectivity during Case 1 increases with decreasing height as quickly as the polarimetric variables change towards a spherical shape dependent signature. Therefore, a strong growth of particle size from pristine shaped particles into almost spherical particles is assumed to be present.

To classify the particle population and the growth process even better the fall streak corrected radar data are analyzed.

The particle growth is visible in the fall streak corrected TARA $sZ$ spectrogram in Figure 6 c). The $sZ$ spectrogram shows a mono-modal particle population where $sZ$ maximum increases from $sZ = -7$ dB$Z$ at the top (3.3 km) to $sZ = 10$ dB$Z$ above the melting layer (indicated by ML). The four Doppler spectra examples in Figure 7 (left column, black spectra) show the evolution from cloud top towards the melting layer. In these examples of Doppler spectra the particle growth is seen

by an increase of spectrum peak values and the broadening of the spectrum width with decreasing height. The fall streak corrected $Z$ profile in Figure 8 a) (black line) shows a 25 dB$Z$ increase of $Z$, from cloud top towards the melting layer. In the profile a slight change in the slope at 3 km is visible. While above 3 km the generation and growth of particles are very fast ($\Delta Z \approx 20$ dB$Z$ in 0.6 km) below the increase is less. There the slope shows a slight increasing linear trend towards the melting layer. Such linearity in the reflectivity profile indicates aggregation of particles (Westbrook et al., 2007). Due to the

homogeneous cloud conditions the contrast of the along fall streak rearranged and vertical $Z$ profiles (light blue profile) is not huge. A larger difference in the data is only visible in the reflectivity of the rain pattern. Therefore, using fall streak rearranged data in Case 1 has its main importance connecting the cloud microphysics with the rain intensity below.

The decrease of the polarimetric radar signatures in Figure 6 e) and f), at N (dotted circles), points out that the ice particles loose their pristine crystal shapes and become spherical particles. Values of $sZ_{\mathrm{DR}}$ decreases in the spectrogram from $sZ_{\mathrm{DR}} \approx$

1 dB near 3.25 km to $sZ_{\mathrm{DR}} \approx 0.2$ dB around 3 km. In the spectrograms it is also visible that the particles at all sizes become spherical, below 3 km the maximum values of $sZ_{\mathrm{DR}} \approx 0.2$ dB. In comparison to the schematic sketch of the $Z_{\mathrm{DR}}$-spectrogram in Figure 3 c) $sZ_{\mathrm{DR}}$ values decrease throughout the whole size range (Doppler velocity range). In Figure 7 this is visible in the column of the spectral differential reflectivity (right column, red spectra). At 3.076 km the $sZ_{\mathrm{DR}}$ signature versus the whole Doppler velocity range in the spectra is close to zero dB. This observation indicates strong aggregation of particles



into spherical ice particles. The fall streak corrected $Z_{\mathrm{DR}}$-profile in Figure 6 d) (black line) shows the strong decrease of polarimetric signature from $Z_{\mathrm{DR}} \approx 1$ dB to $Z_{\mathrm{DR}} \approx 0.1$ dB at 3 km. The high values in the $\rho_{\mathrm{HV}}$ profile in Figure 6 b) exhibit homogeneity of the particles within the measured population. The closer the $\rho_{\mathrm{HV}}$ values are to 1 the more homogeneous are the particle shapes in the sampling volume. $\rho_{\mathrm{HV}}$ values are constantly above 0.9975 below 3 km and therefore match the low

$Z_{\mathrm{DR}}$-values and $sZ_{\mathrm{DR}}$-signatures of a homogeneous spherical particle population.

With TARA operating wavelength it is not possible to have a direct signature related to super-cooled liquid water in the Doppler spectra or bulk parameters, additional information has to be used to detect its presence. The knowledge of the presence of super-cooled liquid water helps in the characterization of the pristine crystal type followed by an even better description of the involved ice crystal growth process. Comparing the observations with the temperature and dew-point-temperature profiles

measured by a radiosonde at 1018 UTC (Figure 9) a super-cooled liquid water layer at cloud top is identified (light blue shaded area). The detected temperature range of $-8°\mathrm{C}$ to $-6\ °\mathrm{C}$ corresponds to the growth regime of needles at that height (Pruppacher and Klett, 1996). As a second indirect validation the observed $sL_{\mathrm{DR}}$ values and signatures in Figure 6 f) are used. They are consistent with other observations and simulated $sL_{\mathrm{DR}}$ values of needle particles (Aydin and Walsh, 1999; Matrosov et al., 2001; Oue et al., 2015b). The simulated $L_{\mathrm{DR}}$ values span a range from $-16$ dB to $-12.5$ dB that is in good agreement

with the observed $sL_{\mathrm{DR}}$ values in the spectrogram that reach values up to $-17$ dB in Figure 6 f). Therefore, the observed signatures of $sL_{\mathrm{DR}}$ and the temperature values indicate the presence of needles at cloud top.

Summing up needles that generated at the cloud top grow fast into a homogeneous population of spherical particles. The fall streak corrected $Z$-profile, Figure 8 a), and the $sZ$-spectrogram in Figure 6 c) show an increase of reflectivity values. The slopes of the $Z$-profile are consistent with the linear increase of reflectivity in case of aggregation of ice particles into snowflakes,

Westbrook et al. (2007). In addition the observed Doppler velocities within Mira radar $sL_{\mathrm{DR}}$-spectrogram of $-2\,\mathrm{m\,s}^{-1}$, right above the melting layer also match the expected velocities for aggregates or slightly rimed particles (Mitchell, 1996; Fukuta and Takahashi, 1999). A strong aggregation or clumping of needle particles is mentioned by Hobbs et al. (1974); Rangno and Hobbs (2001); Hogan et al. (2002). The example of $sZ_{\mathrm{DR}}$-spectra in Figure 7 shows that such spherical particles can be observed within a short distance below particle generation. These compact and dense particles are more efficient to produce

precipitation compared to the ice crystals and snowflakes produced in the time frame before Case 1.

### 6.2   Case 2: generation of a second particle population, 1128-1133 UTC

The focus of Case 2 is the enhanced $Z_{\mathrm{DR}}$ band between 3.2  km and 2.6 km in Figure 4 c) and the related microphysical processes that can be identified and connected to its presence. Figure 4 b) shows that in parallel the reflectivity values increase below 3.2  km, compared to above, which indicates particle growth. Because the band of enhanced $Z_{\mathrm{DR}}$ decreases (particles

become more spherical) towards the melting layer a particle growth process can be assumed in Case 2.

In order to go further in the process description, re-arranged fall streaks are additionally retrieved and analyzed. The retrieval is done setting the initial conditions to 2.25 km for the cloud base height and 90 s for the wind averaging. The $Z_{\mathrm{DR}}$ profile, Figure 10 d) (black), show, at N, the strong increase of the $Z_{\mathrm{DR}}$ profile from $Z_{\mathrm{DR}} \approx 0.2$ dB at 3.2 km to a maximum of 0.65 dB at 2.9 km which indicates the generation of new particles at that height. From N on towards the melting layer, the $Z$ profile,





Figure 8 b) (black), shows a linear increase of reflectivity of $15\,\mathrm{dB}Z$ that indicates an ongoing aggregation process (Westbrook et al., 2007). The rearranged $Z_{\mathrm{DR}}$ profile in Figure 10 d) shows also signatures that are in agreement with an aggregation process. Below N, the $Z_{\mathrm{DR}}$-values decrease due to the aggregation of the new generated particles into more spherical shaped particles. It is worth noting that such signatures can not be identified in the vertical, not re-arranged, $Z$ and $Z_{\mathrm{DR}}$ profiles

(light blue profiles in the Figures 8 b) and 10 d)). This analysis therefore demonstrates the advantage of using along fall streak rearranged radar data, as obtained from Pfitzenmaier et al. (2017). Compared to Case 1, the slope of the $Z$-profile in Case 2 increases much slower at a rate of $15\,\mathrm{dB}Z$ per $1\mathrm{km}$ (in Case 1 $25\,\mathrm{dB}Z$ per $1\mathrm{km}$). The different slopes of the $Z$-profiles might indicate different aggregation processes for the two cases. A reason could be that in Case 2 particles seed from above which is not the case in Case 1, where particles are directly generated at cloud top.

In Figure 10 b) the $\rho_{\mathrm{HV}}$-minimum of $0.956$ right above the increase of $Z_{\mathrm{DR}}$ at $3.2\mathrm{km}$ indicates the generation of new particles. Due to a high variation of particle shapes (new generated particles, and seeded particles) within the sampling volume the values of $\rho_{\mathrm{HV}}$ lead to that minimum. Below $3.1$ km, the $\rho_{\mathrm{HV}}$ value increases and reaches values $> 0.99$ below $2.8$ km. This increase of $\rho_{\mathrm{HV}}$ shows that the particle population becomes more and more homogeneous which indicates an aggregation process.

Next, rearranged fall streak data are analyzed in the spectral domain in order to discriminate between the generated particles at N and the particles seeding from above. The rearranged $sZ$ and $sZ_{\mathrm{DR}}$ spectrograms are presented in Figure 10 c) and e). At $3.1$ km, a broadening is observed in the $sZ$ spectrogram. This broadening of the spectra corresponds to the generation of a second particle mode that is visible a N. The maximum $sZ$ values increase from $sZ \approx -11$ dB$Z$ above N to $sZ \approx 7$ dB$Z$ right above the melting layer. These observations are indicating aggregation of the newly generated crystals towards

the melting layer. In Figure 10 e) a bimodality is visible in the $sZ_{\mathrm{DR}}$-spectrogram at N. In comparison to the $sZ$ spectrogram the $sZ_{\mathrm{DR}}$ spectrogram contains a bimodal shape from N till about $350$ m lower. In the $sZ$-spectrogram such separated modes cannot be identified. Because of the stable second mode in the $sZ_{\mathrm{DR}}$ spectrogram the assumption of an aggregation process below N is adjusted and a separated diffusional growth of the new generated particles can be assumed before they aggregate with the ice particles seeded from above.

The analysis of the single $sZ$ and $sZ_{\mathrm{DR}}$ spectra in Figure 11 confirms the hypothesis of diffusional growth. Figure 11 displays four fall streak rearranged Doppler spectra ($sZ$, left column, black spectra) and differential reflectivity spectra ($sZ_{\mathrm{DR}}$, right column, red line) at four different heights, $3394$ m, $3055$ m, $2864$ m, and $2630$ m. At N, a clear broadening in the $sZ$ spectra is visible that is caused by the development of a second particle mode. With decreasing height, the shape of $sZ$ is again monomodal and its values rise, see $sZ$ at $2864$ m, and $2630$ m. At $3055$ m, it is seen that the newly developed particles

at N have high $sZ_{\mathrm{DR}}$-values and the $sZ_{\mathrm{DR}}$ spectrum shape turns bimodal. However, in comparison to the $sZ$ that looses its bimodal shape rather quickly the $sZ_{\mathrm{DR}}$ maintains it further below. At $2864$ m still two particle populations can be identified in the $sZ_{\mathrm{DR}}$ spectrum. Between $-8.5\,\mathrm{m\,s^{-1}}$ and $-7.25\,\mathrm{m\,s^{-1}}$, high $sZ_{\mathrm{DR}}$-values indicate a large amount of pristine shaped ice crystals, while at Doppler velocities $< -8.5\,\mathrm{m\,s^{-1}}$ low $sZ_{\mathrm{DR}}$-values refer to almost spherical crystals in that part of the spectrum. This clearly shows that from N to $2864$ m the generated particles keep their shape dependence and, therefore, can



be separated using the $sZ_{\mathrm{DR}}$ spectrum. In parallel the $sZ$ values increases for Doppler velocities $> -8.5\,\mathrm{m\,s^{-1}}$. Due to this growth $sZ$ looses a clearly visible bimodality that was present at N.

These observed signatures fit to the schematic sketches of the diffusional growth of particles, Figure 3 b). The signatures of the $sZ$ and the $sZ_{\mathrm{DR}}$ clearly show that particles generated at N grow separately from the particles seeding from above. Due to the clear size dependent growth of the smaller particles we assume a diffusional growth of the newly generated particles. Nevertheless, at lower heights it is seen that the clear separation in the $sZ_{\mathrm{DR}}$ spectra disappears. This can be explained by the merging of the two particle populations and, therefore, an aggregation process between the particles, see spectra at 2630 m. At N and at 2864 m, the $sZ_{\mathrm{DR}}$-spectrogram shows negative signatures for large ice particles. The values show that the growth at N leads to prolate oriented particles. Super cooled liquid water droplets can lead to partly riming of large particles or turbulence could lead to this prolate orientation. However, with decreasing height the shape of these particle populations becomes spherical or even slightly oblate again. Nevertheless, the analysis of the along fall streak rearranged spectral data has demonstrated that such processes cannot be identified using integrated volume data. Because using integrated moment data such size dependent spectral signatures are not longer present.

The particle type of the new generated particles can be identified by combining the polarimetric measurements from TARA with additional data from the radiosonde measurements and the $sL_{\mathrm{DR}}$-signatures from the Mira radar. Bands of enhanced $Z_{\mathrm{DR}}$ are an indicator for the generation and growth of pristine ice particles, mainly dendrites or hexagonal plates (Bader et al., 1987; Kennedy and Rutledge, 2011; Moisseev et al., 2015). However, the visible polarimetric signatures in the radar measurements in Case 2 are caused most likely by newly generated ice needles or columns. The temperature ranges of the radiosonde launches in Figure 9 show values between $-8^{\circ}\mathrm{C}$ and $-5^{\circ}\mathrm{C}$ at 3 km for both launches. This temperature corresponds to a needle or column generation regime, as in Case 1. In addition the $sL_{\mathrm{DR}}$-spectrogram shows similar signatures with $sL_{\mathrm{DR}}$-values of $-17$ dB that agree to the simulation signatures for needles or columns (Aydin and Walsh, 1999; Matrosov et al., 2001; Oue et al., 2015b). Therefore, it is expected that the enhanced $Z_{\mathrm{DR}}$ signature is caused by needles or columns. An indication for needle particles can be the presence of a supercooled liquid water layer at that height. Hogan et al. (2002) showed that small-scale dynamics within frontal system clouds lead to super-cooled liquid droplet formation in the cloud system. Such upward motions are visible in the vertical Doppler velocity field during Case 2 in Figure 5, and the hypothesis of needle generation is considered. Nevertheless, in comparison to Case 1, the growth process of the particles is different.

Finally, it is observed, that the new generated particles at 3.1 km and the related growth of ice crystals lead to an increase of $Z$ in the rain pattern below the melting layer. The radiosonde temperature profiles and updraft patterns in the vertical Doppler velocity field show the potentiality for supercooled liquid water layer presence around 3 km. The measured $sL_{\mathrm{DR}}$-signatures show signatures of needle or columnar shaped ice particles. Therefore, the existence of a supercooled liquid water layer is possible. Furthermore, the $sZ_{\mathrm{DR}}$-spectrogram clearly exhibits two different particle populations below this height. The decrease of polarimetric signature in parallel to the increase of $Z$-values suggest the aggregation of the two present particle populations during Case 2. Deeper analysis of the $sZ_{\mathrm{DR}}$ spectra indicate that the mode of smaller particles first grows due to the diffusional growth before merging with the other mode. When these two modes merge, the clear separation disappears and the aggregation of the particles becomes dominant.





### 6.3 Case 3: Particle growth of hexagonal particles, 1135-1147 UTC

Figure 12 shows the analysis of the retrieved fall streak from 114730 UTC. The focus of Case 3 is related to the increased $Z_{DR}$ signature at $5\,\mathrm{km}$ between 1135 UTC and 1147 UTC. At that height the $Z$-values increase and indicate a particle growth. The fall streak rearranged data are retrieved using a cloud base height of $2.25\,\mathrm{km}$ and a $90\,\mathrm{s}$ averaged wind profile. The vertical

profiles and fall streak corrected data are only analyzed from cloud top till $3.0\,\mathrm{km}$. This is done because at $3.0\,\mathrm{km}$ a strong shear in wind direction is visible in Figure 5 d). Therefore, a homogeneous cloud cannot longer be assumed and the along the fall streak rearranged data would describe a different particle population below $3.0\,\mathrm{km}$. So the analysis focuses at regions above $3.0\,\mathrm{km}$ and no link to the increased precipitation pattern is done for that case. The case is anyhow challenging. Because the data above $3.0\,\mathrm{km}$ show signatures of aggregation and some indications for riming are found. Therefore, the focus of Case 3

is the identification of the most probable particle growth process.

Figure 4 b) depicts a horizontal homogeneous $Z$-field during Case 3 where the main increase of reflectivity is visible between $5\,\mathrm{km}$ and $4.5\,\mathrm{km}$. In the fall streak corrected $sZ$ spectrogram, Figure 12 c) and Figure 13, the increase of $sZ$ can be localized at the same height range. There the maximum values of $sZ$ increase from $-5\mathrm{dB}Z$ above $5\,\mathrm{km}$ to $sZ \approx 6\ \mathrm{dB}Z$ at around $4.5\,\mathrm{km}$ ($sZ \approx 7.5\ \mathrm{dB}Z$ at $4\,\mathrm{km}$). The $Z$-profile in Figure 12 d) (black line) shows constant values above $5\,\mathrm{km}$ ($Z \approx 10\ \mathrm{dB}Z$)

and below $4\,\mathrm{km}$ ($Z \approx 20\ \mathrm{dB}Z$). The main growth process of the tracked particle population is visible between $5\,\mathrm{km}$ and $4\,\mathrm{km}$, where the reflectivity increases linearly of $10\ \mathrm{dB}Z$, which indicates an ongoing aggregation (Westbrook et al., 2007). This signature is consistent with other observations and, therefore, the studies of the microphysics are based on fall streak rearranged TARA data.

The $sZ_{DR}$ spectrogram exhibits a clear shape-size dependency even below the particle growth. The $sZ_{DR}$ signature consists

of high values at the right edge where the small particles are present. This signature is constantly present between $4.5\,\mathrm{km}$ and $3\,\mathrm{km}$. The maximum values of $sZ_{DR}$ are just below $5\,\mathrm{km}$, $sZ_{DR} \approx 2.25\ \mathrm{dB}$, $sZ_{DR}$-spectra at P in Figure 13, and decrease with the height. Nevertheless, the shape-size dependency stays. The fall streak corrected $Z_{DR}$ profile in Figure 12 f) exhibits a maximum of $0.75\ \mathrm{dB}$ between $5.0\,\mathrm{km}$ and $4.5\,\mathrm{km}$. Below $Z_{DR}$ values are rather constant around $0.4\ \mathrm{dB}$ till $3.0\,\mathrm{km}$. Such a bulk volume is challenging to interpret without the spectral polarimetric measurement.

The profile of $\rho_{HV}$ in Figure 12 b) increases with the observed particle growth below $5\,\mathrm{km}$ ($Z$ profile) and stays constant from $4.5\,\mathrm{km}$ to $3\,\mathrm{km}$. It also exhibits a minimum before the increase around $5\,\mathrm{km}$ (slightly above the maximum of $Z_{DR}$). This is caused by a large variety of particle shapes before the growth, like in Case 2. Below the growth process $\rho_{HV}$ shows a constant value that is $0.05$ lower than in Case 2. This is caused by the shape-size relation that is present in the $sZ_{DR}$ spectrogram. Because this signature stays constant till $3\,\mathrm{km}$ further growth processes of the tracked particle population like aggregation do

not occur.

The constant shape-size dependency within the $sZ_{DR}$ spectrogram between $4.5\,\mathrm{km}$ and $3.0\,\mathrm{km}$ is the most dominant feature. Comparing these signatures to the schematic sketches in Figure 3 we can opt for riming or aggregation of particles. In Figure 13 the Doppler spectra do not show a separated mode. Simulations of $sZ_{DR}$ signature indicate that the observed signature at P in Figure 13, $4.794\,\mathrm{km}$, can be produced by a mixture of pristine hexagonal shaped ice particles and aggregated or rimed particles





(Spek et al., 2008). Like in Case 1 and 2, the trend of the $sZ_{DR}$ versus Doppler velocity can be examined, when it is stable for several heights and times. However, caution is required for the values of $sZ_{DR}$ because of their large variance. The absolute values of the Doppler velocities of the Mira spectra (not shown) are below $2\,\mathrm{m\,s^{-1}}$. They correspond to pristine and slightly rimed particles (Mitchell, 1996). Due to the homogeneity of the particle population below $4.5\,\mathrm{km}$ in terms of $sZ$ and $sZ_{DR}$ we

assume that the particles mainly grow at P and then seed further through the cloud system. Below $4\,\mathrm{km}$ the $sZ_{DR}$ spectrogram and spectra in Figure 12 e) and Figure 13 (right column) show that the larger particles could be prolate. Negative differential reflectivity signatures can be expected for conical graupel presence (Oue et al., 2015a). However, the observations of Oue et al. (2015a) indicated that the reflectivity of graupel particles is much higher than in our observations. Riming of particles that lead to prolate particles is also possible but the presence of conical graupel cannot be confirmed. Nevertheless, the observed vertical

Doppler velocities of the Mira radar are within the expected range for slightly rimed particles. In comparison to the schematic sketches in Figure 3 a) no separated particle mode is visible. However, the Doppler spectra at $4031\,\mathrm{m}$, and $3309\,\mathrm{m}$ in Figure 13 exhibit a small bump where $sZ_{DR}$ signatures are negative. Therefore, a small amount of rimed prolate oriented particles could be present.

The identification of the dominant particle shape at $5\,\mathrm{km}$ can indicate the dominant growth process. Because dendrites

and plates have different efficiencies of aggregation and riming (Pruppacher and Klett, 1996; Lamb and Verlinde, 2011), the temperature of radiosonde launches is used to identify the possible pristine particle growth regime. In Figure 9 both profiles indicate a temperature range between $-15°$ and $-13°$ C, but a direct identification of a super-cooled liquid water layer around $5.0\,\mathrm{km}$ cannot be done (dew-point-temperature and temperature profiles do not match at that height). Nevertheless, the temperature range at $5.0\,\mathrm{km}$ agrees with the growth regime of hexagonal plates or dendrites (Pruppacher and Klett, 1996; Fukuta

and Takahashi, 1999). As in Case 2, the vertical Doppler velocity measurements indicate updraft patterns in that region that increases the probability of the presence of super-cooled liquid water droplets (Hogan et al., 2002). However, from the observations it is not clear which pristine particle shape is dominating the growth process at that height: hexagonal plates or dendrites. Summing up all the facts, the polarimetric measurements, the temperature range, and the updraft patterns we conclude that the presence of supercooled liquid water around P is possible. Because no separated riming mode could be identified in the spectra

we do not expect riming to be the dominant particle growth process. It is more likely that the constant reflectivity values and the constant shape-size relationship along the fall streak rearranged data below P are caused by an aggregation process. However, more measurements of the particle microphysics would be necessary to get a more detailed picture on which growth process might dominate in that case.

## 7   Discussion and conclusion

The high temporal and spatial resolution of profiling S-band radar TARA made it possible to detect and identify structures of particle generation and growth within complicated and dynamically complex mixed-phase cloud systems. Using a S-band radar system has the advantage that the attenuation of the radar signal during precipitation can be neglected compared to higher frequencies operating systems (Kollias et al., 2007). The unique 3 beam configuration of TARA makes it possible to retrieve



the full 3 D wind field for each measurement (Unal et al., 2012). Furthermore, this dynamical information is used to retrieve particle fall-streaks within the bulk parameter fields, Pfitzenmaier et al. (2017). Therefore, the bulk parameter profiles, $Z$, $Z_{\mathrm{DR}}$, and $\rho_{HV}$, as well as spectrograms, $sZ$ and $sZ_{\mathrm{DR}}$, are rearranged along such retrieved fall streak before they are analyzed. Consequently, these corrected spectrograms and profiles show the consistency of the particle growth within the cloud part and relate better to the increase of the rain pattern below the melting layer. Such increased rain patterns are identified during Case 1 and 2.

The analysis of the along fall streak rearranged measurements allows to localize the particle growth process. This can be done by examination of the increase and the slope of $Z$ along the fall streak. Westbrook et al. (2007) showed that the aggregation of ice crystals leads to a constant increase of Z with decreasing height. The observations were based on averaged data of homogeneous ice clouds over a long time period. Using rearranged data along fall streak such studies can be done in less homogeneous cloud conditions. The presence of super-cooled liquid water leads to a much faster growth of ice particles within a shorter amount of time (fast increase of $Z$ within short height ranges) compared to the growth of particles by aggregation in the ice phase only (Pruppacher and Klett, 1996). That may explain the increase in the $Z$ profile within 1 km of 25 dB$Z$ in Case 1, 15 dB$Z$ in Case 2, and 10 dB$Z$ in Case 3.

The fall streak corrected $sZ_{\mathrm{DR}}$ measurements of TARA are able to deliver shape information and their distribution within the tracked particle population. The changes of the observed signature is in good agreement with the discussed particle growth (increase of $Z$). However, stable trends of $sZ_{\mathrm{DR}}$ should be examined, rather than absolute values because of the large variance of this spectral polarimetric parameter.

In Case 3 the observed polarimetric signature in the $sZ_{\mathrm{DR}}$ spectrogram agrees with the simulated $sZ_{\mathrm{DR}}$ signature for a mixture of hexagonal plates and rimed particles or aggregates in Spek et al. (2008). Therefore, we can opt for riming and/or aggregation of particles in that case. Aggregation is assumed to be the dominant particle growth process. Nevertheless, negative $sZ_{\mathrm{DR}}$-values are observed in the spectrogram, which may indicate rimed prolate oriented particles. However, a typical separated riming mode within the Doppler spectra $sZ$ is not visible. The Mira vertical Doppler velocities fit to the velocity range of slightly rimed particles. Observed updraft patterns indicate the possible presence of supercooled liquid water at the region of the main particle growth, like observed by (Hogan et al., 2002). Therefore, riming cannot be neglected in Case 3, however, explicit riming signatures cannot be observed in the measured spectra.

The observed signatures of $sL_{\mathrm{DR}}$ and $sZ_{\mathrm{DR}}$ as well as the measured temperature in Case 1 are in agreement with modeled signatures of needles and their strong tendency to clump and aggregate as reported by (Rangno and Hobbs, 2001). In Case 2 the observed signatures are more complex. While the temperature range can be considered the same as Case 1 the polarimetric signature differs. One reason can be the difference in aggregation efficiency of the new produced particles in these two cases. In Case 2 the needles show a lower aggregation efficiency with the ice particles seeded from above, than in Case 1 where only needles were present. The decrease of the polarimetric signature is less strong than in Case 1. Another reason might be a difference in the particle concentration of the generated particles that is less in Case 2 ($sZ$ of the increased $sZ_{\mathrm{DR}}$ mode is less that the $sZ$ values in Case 1). Then the aggregation efficiency is less and the decrease of the $sZ_{\mathrm{DR}}$ signature takes longer. Nevertheless, from the observations it is not visible which mechanism is dominant or if they depend on each other. It





is only seen that a second population of particles with a high shape size dependency is present in the data. These polarimetric signatures as well as the corresponding temperature profiles indicate the presence of super-cooled liquid water.

Parallel radiosonde launches and the $sL_{DR}$ of the vertical pointing $K_a$ radar Mira are used to compare and verify the results based on the TARA measurements. The cases presented in Section 6 are examples of the spectral and bulk parameter

characteristics observed during the whole event. These analyzed ice particle growth processes above the melting layer show a strong correlation with increased pattern of precipitation in the reflectivity fields. Therefore, the presented data with applied fall streak correction demonstrate the advantage of using also high resolution S-band radar data to increase the understanding of ice particle growth within precipitating mixed-phase cloud systems.

*Author contributions.* C. Unal designed the data processing of TARA, contributed to the quality control of the data and the result discussion.

Y. Dufournet helped organizing and performing the ACCEPT campaign and discussing the results. H. Russchenberg designed the scientific structure of the project, helped in the realization of ACCEPT, and contributed to the discussion of the results. L. Pfitzenmaier performed the measurements taken during the ACCEPT campaign in autumn 2014, Cabauw, the Netherlands, supported by the colleagues from TROPOS. L. Pfitzenmaier performed the fall streak analysis, contributed to the result discussion, and prepared the manuscript with contributions of all coauthors.

*Competing interests.* The authors declare that they have no conflict of interest.

*Acknowledgements.* The research leading to these results has received funding from the European Union Seventh Framework Program (FP7/2007−2013): People, ITN Marie Curie Actions Program (2012−2016) in the frame of ITaRS under grant agreement no. 289923. The ACCEPT campaign was partly founded by ACTRIS Research Infrastructure Project by the European Union's Horizon 2020 research and innovation program under grant agreement no. 654169 and previously under grant agreement no. 262254 in the Seventh Framework Program

(FP7/2007−2013). Authors want to acknowledge also the cooperations of institutes (TROPOS, KNMI, LMU) and companies (METEK GmbH) during ACCEPT. We highly appreciated the discussions at EGU 2015 at the Poster session that inspired the presented work.





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





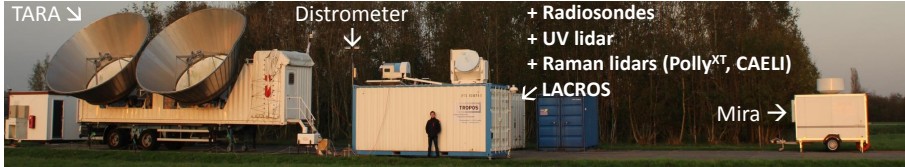

**Figure 1.** Measurement setup of the ACCEPT campaign at CESAR. The arrows point on the TARA radar (radar on the left), the disdrometer that is mounted on the LACROS container, and the vertical pointing Mira cloud radar (35 GHz). Complementary instruments of ACCEPT are listed but not visible on this photo.

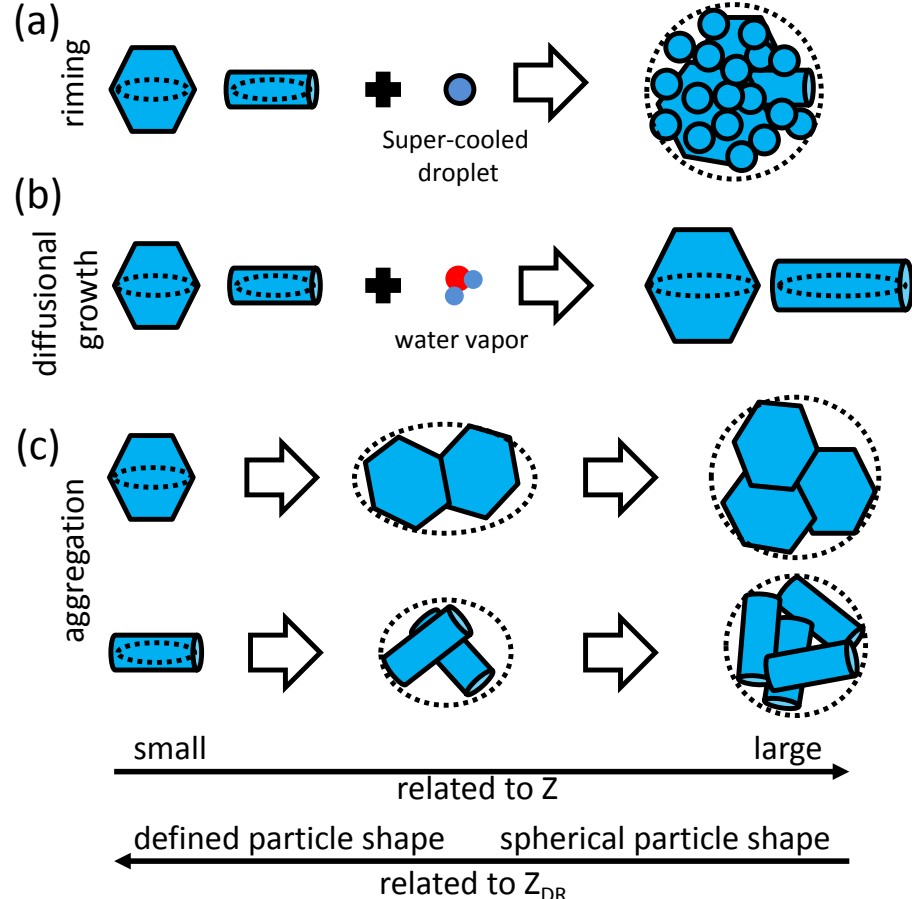

**Figure 2.** Sketch of ice particle growth processes within mixed-phase clouds. In a) riming of ice crystals is depicted. b) shows the diffusional growth of ice crystals and c) the aggregation of ice crystals. The hexagonal plates and cylindrical columns represent either plate and dendrite shape or needles and columns, respectively. Size growth is indicated with bigger particles in the sketch. In the measurements an increase in particle size is related to an increase in $Z$ values, indicated by the first arrow at the bottom. The particle shapes are indicated by the dotted ellipses (the particles are modeled as spheroids). The second arrow at the bottom indicates the decrease of $Z_{DR}$ as an indicator for the decrease of defined particle shape with increasing size.




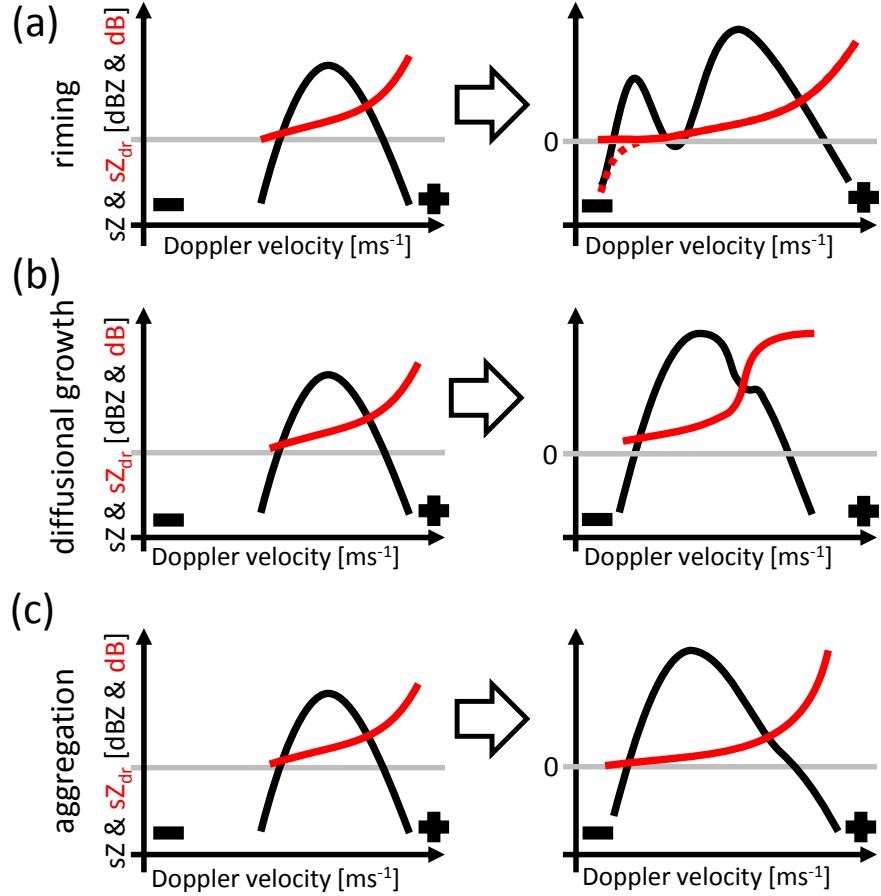

**Figure 3.** Sketch of signature of (a) riming, (b) diffusional growth, and (c) aggregation of ice particles and their corresponding changes in the Doppler spectra $sZ$ (black lines) and spectral differential reflectivity $sZ_{DR}$ (red lines). All sketches represent theoretical dependencies, the gray line indicates 0 dB for the $sZ_{DR}$ values. The Doppler velocity values are relative values (left: minus sign: large negative Doppler velocities; right: plus sign; very small or positive Doppler velocities; negative Doppler velocities indicate movements towards the radar). Conceptional sketches are based on spectral simulations of Spek et al. (2008); Dufournet and Russchenberg (2011); Oue et al. (2015a).





**Figure 4.** Overview of the radar measurements used in this paper. b) and c) show reflectivity and differential reflectivity measurements, obtained with TARA radar (S-band). a) and d) show the reflectivity and linear depolarization ratio measurements from the Mira radar ($K_a$-band). The boxes 1 , 2, and 3 highlight the time frames where fall streaks at 101250 UTC, 113331 UTC and 114730 UTC are retrieved and their rearranged data are discussed and analyzed



**Figure 5.** Overview of the dynamic variables measured and retrieved by TARA radar (S-band). In a) the Doppler width is displayed, b) shows the retrieved horizontal wind velocity, c) the retrieved vertical Doppler velocity, and d) the retrieved wind direction. The boxes 1, 2, and, 3 highlight the time frames where fall streaks at 101250 UTC, 113331 UTC and 114730 UTC are retrieved, and their rearranged data are discussed and analyzed.





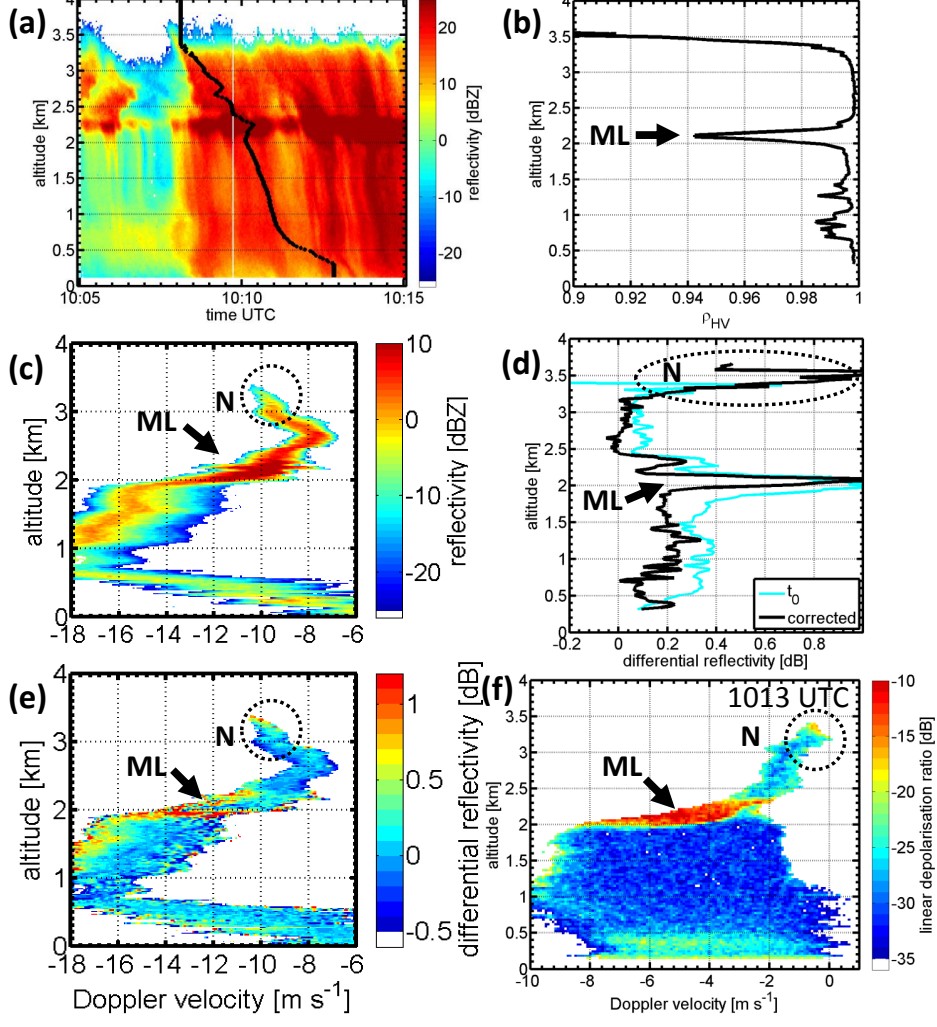

**Figure 6.** a) shows the retrieved fall streak at 101250 UTC obtained with TARA. c) and e) are the fall streak corrected spectrograms ($sZ$ and $sZ_{DR}$, $45°$ elevation - Note that the Doppler velocity contains the radial wind). b) and d) show the fall streak corrected profiles of $\rho_{HV}$ and $Z_{DR}$ in black while light blue represents the vertical $Z_{DR}$ profile at 101250 UTC. f) shows the $sL_{DR}$ spectrogram of the vertical pointing Mira at 1013 UTC (all data displayed in the spectrograms have $SNR > 10$ dB). ML points out the signature of the melting layer and N the signature related to the needle particles.

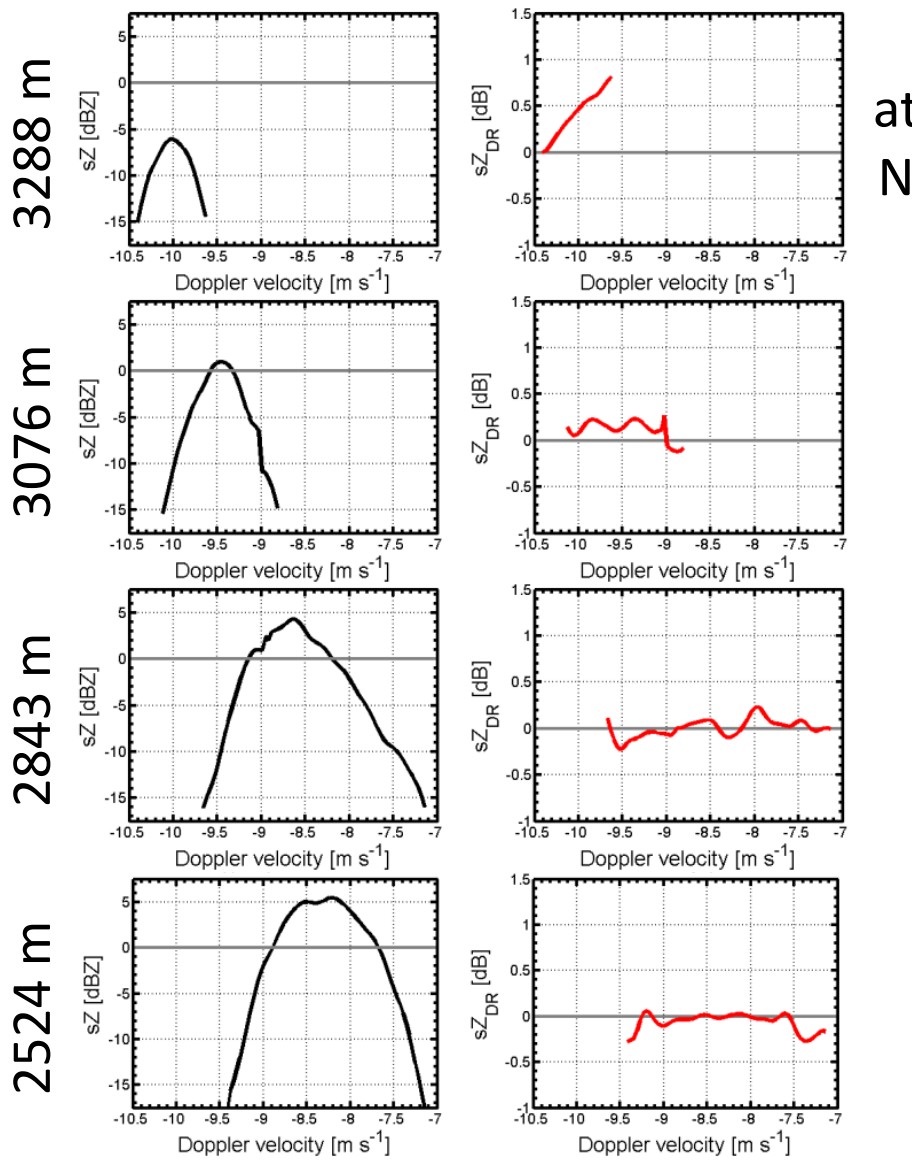

**Figure 7.** Along the fall streak at 101250 UTC rearranged $sZ$ and $sZ_{\mathrm{DR}}$ at four different altitudes. The left panel shows Doppler spectra at 3288 m, 3076 m, 2843 m, and 2524 m. The right column displays the corresponding spectral differential reflectivity at the same heights. All spectra are at $45°$ elevation and averaged over 3 consecutive time bins.



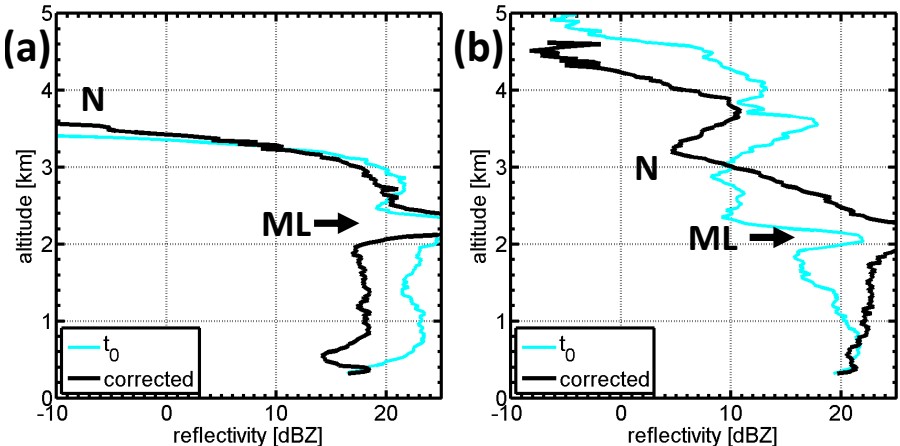

**Figure 8.** a) and b) show the fall streak corrected profiles of $Z$. a) shows the profiles at 101251 UTC and b) at 113331 UTC. The black line is the fall streak corrected profile and the light blue line represents the vertical profile at the given time. ML points out the signature of the melting layer and N the signature related to needle growth and new generated particle population, respectively.

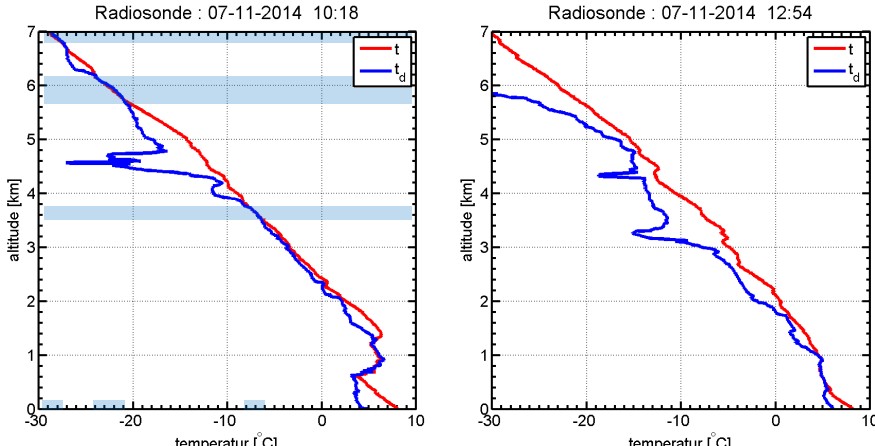

**Figure 9.** Radiosonde launches from the CESAR site at 1018 UTC (left) and after precipitation system has passed at 1254 UTC (right). Displayed are profiles of the temperature $t$ (red) and dew-point temperature $t_d$ (blue). The light blue bars indicate areas where supercooled liquid water was detected with the radiosondes, their corresponding temperature range on the x-axis is highlighted.





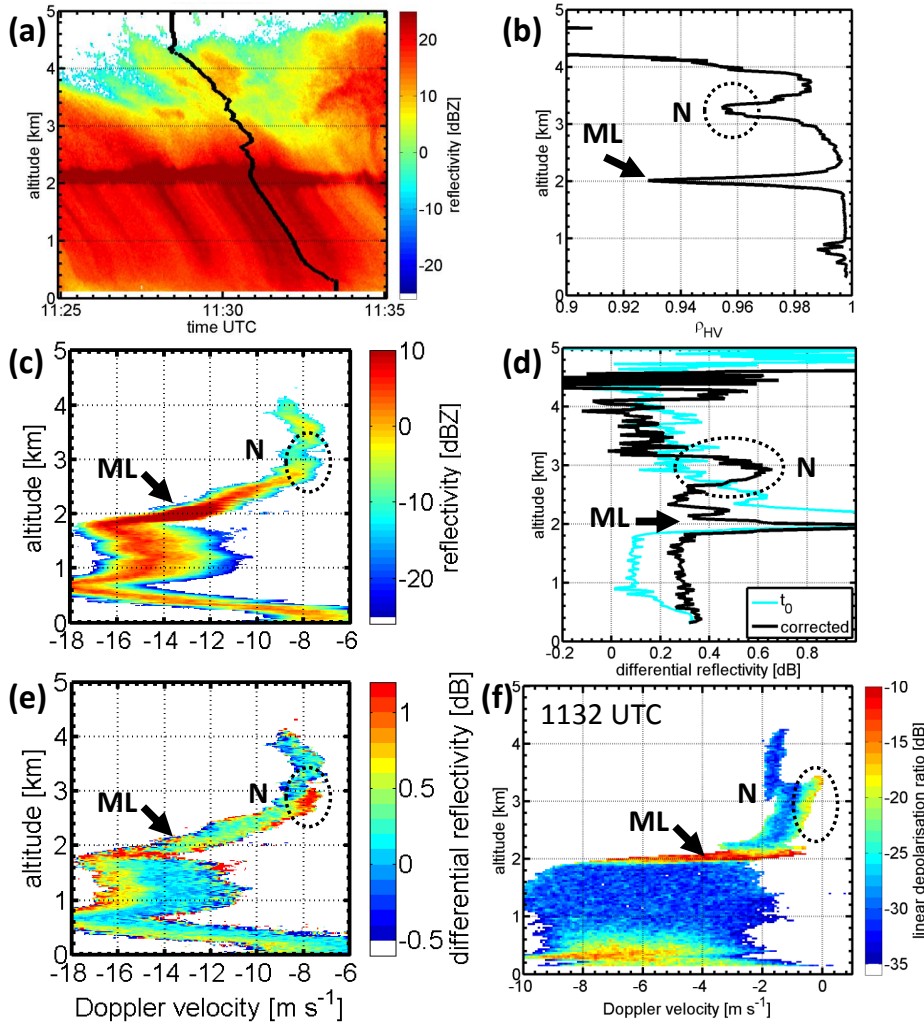

**Figure 10.** a) shows the retrieved fall streak at 113331 UTC on top of the $Z$ field. c) and e) are the fall streak corrected spectrograms ($sZ$ and $sZ_{DR}$, 45° elevation - Note that the Doppler velocity contains the radial wind). b) and d) show the fall streak corrected profiles of $\rho_{HV}$ and $Z_{DR}$ in black while light blue represents the vertical $Z_{DR}$ profile at 113331 UTC. f) shows the $sL_{DR}$ spectrogram of the vertical pointing Mira at 1132 UTC (all data displayed in the spectrograms have $SNR > 10$ dB). ML points out the signature of the melting layer and N the signature related to the new particle population.



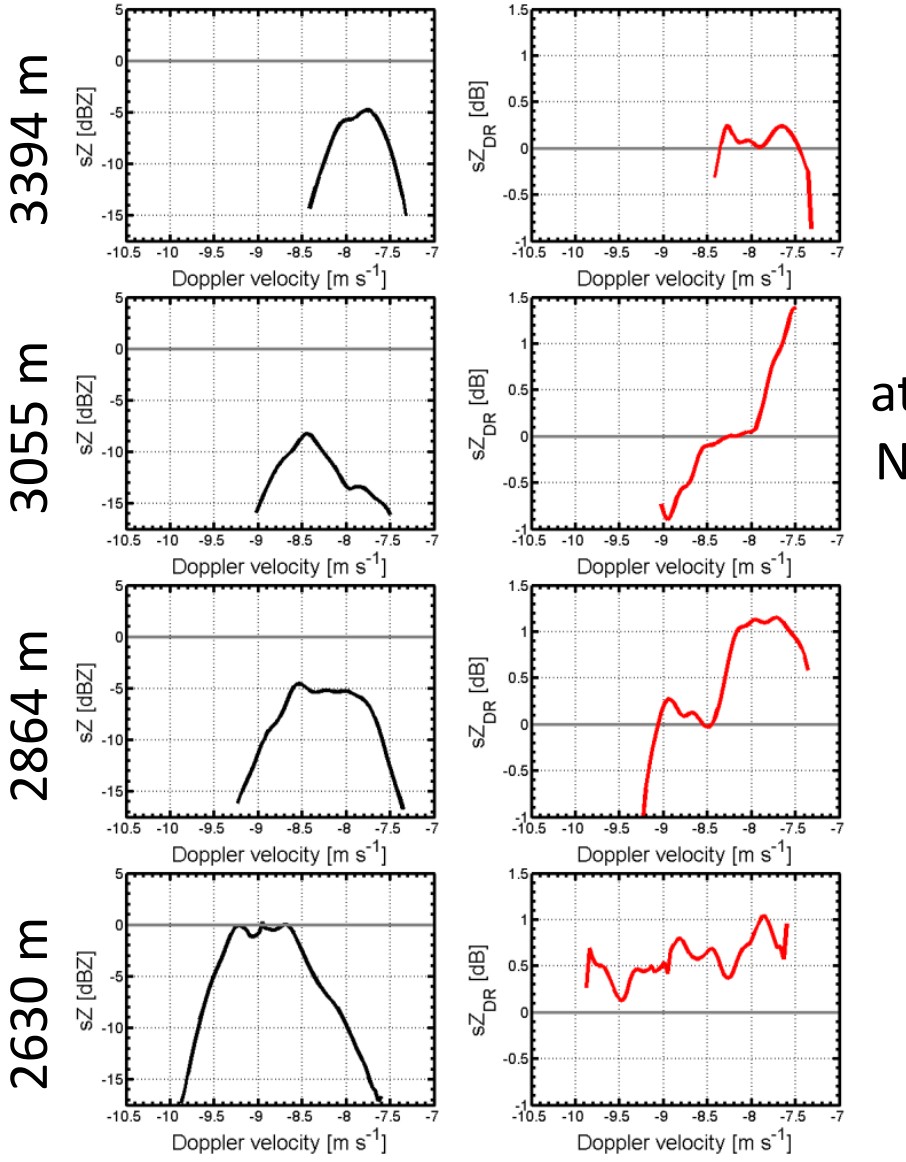

**Figure 11.** Along the fall streak at 113331 UTC rearranged $sZ$ and $sZ_{DR}$ at four different altitudes. The left panel shows Doppler spectra at 3394 m, 3055 m, 2864 m, and 2630 m. The right column displays the corresponding spectral differential reflectivity at the same heights. All spectra are at 45° elevation and averaged over 3 consecutive time bins.



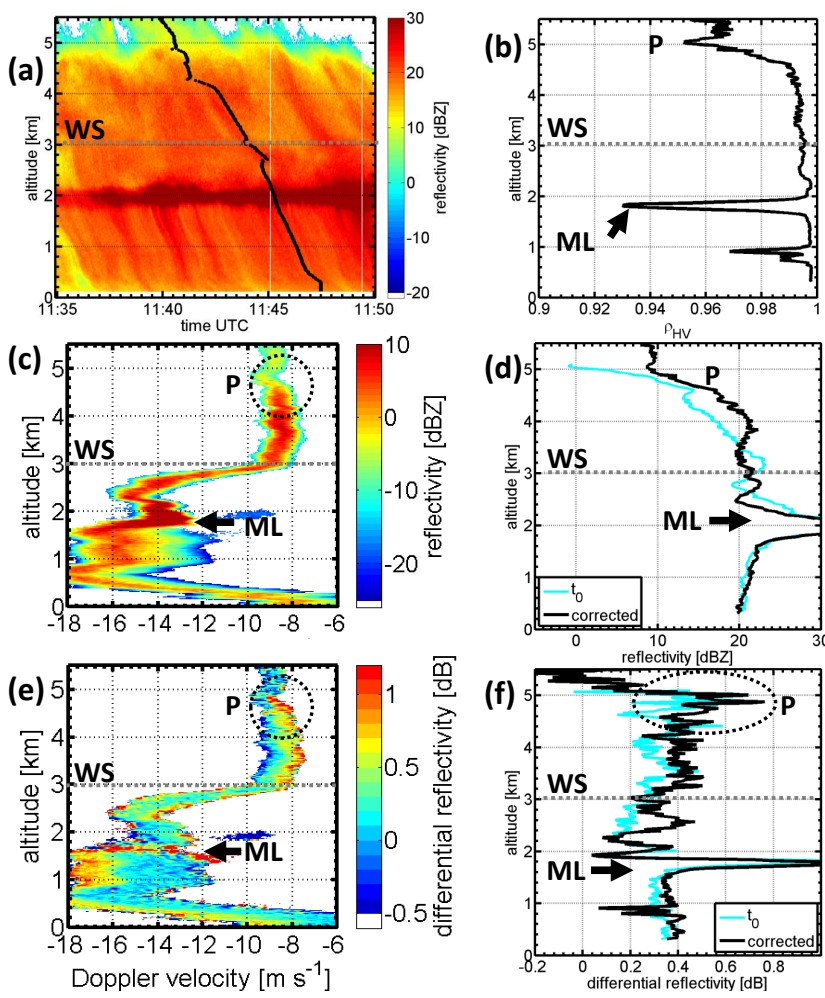

**Figure 12.** a) shows the retrieved fall streak at 114730 UTC obtained with TARA. c) and e) are the fall streak corrected spectrograms ($sZ$ and $sZ_{\mathrm{DR}}$, $45°$ elevation , all data displayed in the spectrograms have $SNR > 10$ dB - Note that the Doppler velocity contains the radial wind). b), d) and f) show the fall streak corrected profiles of $\rho_{\mathrm{HV}}$, $Z$, and $Z_{\mathrm{DR}}$ in black while the lines in light blue represent the vertical $Z$ and $Z_{\mathrm{DR}}$ profiles at 114730 UTC. ML points out the signature of the melting layer and P (black dotted circles) the signature related to the growth process of the particle population under investigation. The dotted line WS indicates approximately the wind direction shear height.

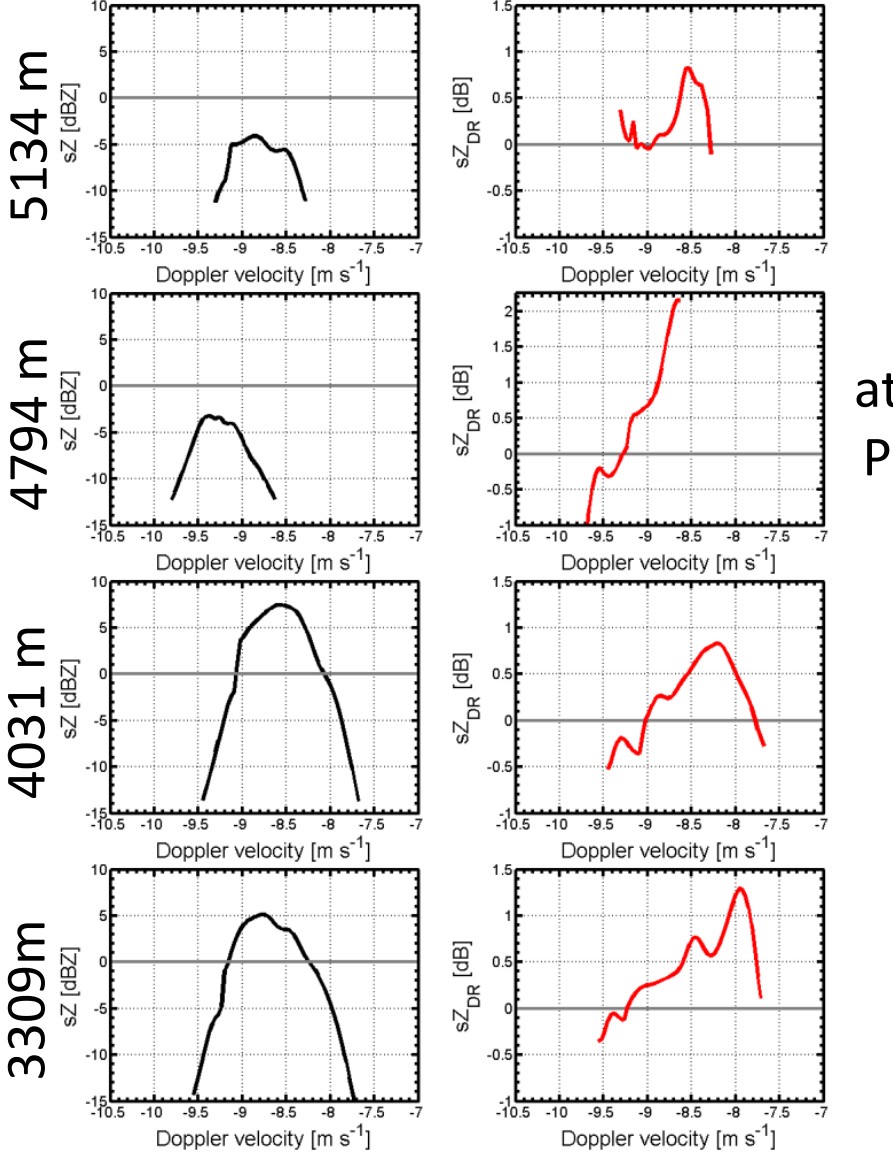

**Figure 13.** Along the fall streak at 114730 UTC rearranged $sZ$ and $sZ_{\mathrm{DR}}$ at four different altitudes. The left panel shows Doppler spectra at 5134 m, 4794 m, 4031 m, and 3309 m. The right column displays the corresponding spectral differential reflectivity at the same heights. All spectra are at $45°$ elevation and averaged over 3 consecutive time bins.



**Table 1.** Specifications of the TARA radar during the ACCEPT campaign.

| Radar | | |
|---|---|---|
| Type | FM-CW | |
| Central frequency | 3.298 GHz | S-band |
| Transmitted power | 100 W | Automatic decrease by step of 10 dB in case of receiver saturation |
| **Signal generation** | | |
| Sweep time | 0.5 ms | |
| No. of range bins | 512 | |
| Range resolution | 30 m | Height resolution = 21.2 m |
| Time resolution | 2.56 s | |
| **Polarimetry** | | |
| Polarisation | VV HV HH | Main beam only |
| Measurement cycle | VV HV HH OB1 OB2 | Main beam + 2 offset beams |
| **Doppler** | | |
| No. Doppler bins | 512 | |
| Doppler resolution | $0.036 \, \mathrm{m\,s^{-1}}$ | |
| Max. unambiguous vel. | $\pm 9.1 \, \mathrm{m\,s^{-1}}$ | |
| Max. vel. main beam | $\pm 45.5 \, \mathrm{m\,s^{-1}}$ | After spectral polarimetric dealiasing (Unal and Moisseev, 2004) |
| Max. vel. offset beams | $\pm 45.5 \, \mathrm{m\,s^{-1}}$ | After spectral dealiasing |
| **Antennas** | | |
| Beam width | $2.1°$ | |
| Gain | 38.8 dB | |
| Near field | $\leq 200$ m | |
| **Beams** | Elevation | Azimuth (North= $0°$) |
| Main beam | $45°$ | $246.5°$ |
| Offset beam 1 | $60°$ | $246.5°$ |
| Offset beam 2 | $43.1°$ | $267.3°$ |
| **Clutter suppression** | | |
| Hardware | Antennas | Low side lobes |
| Processing | Doppler spectrum | Spectral polarimetry (main beam) |