# Peer review of "Observing ice particle growth along fall streaks in mixed-phase clouds using spectral polarimetric radar data"

_Atmospheric Chemistry and Physics, 2017_

## Referee Comment (RC1) · Anonymous Referee #2 · 25 Feb 2018

General Comment: This study collected polarimetric Doppler spectra at an elevation angle of 45 degrees. The observed spectra were realigned with height along with retrieved fall streaks and analyzed the reflectivity and differential reflectivity spectra changing with height to discuss ice particle growth. The novelty technique and idea used in this study are very interesting, but sometimes I was confused by increase/decrease of fall speed when looking at the observed spectra. Because horizontal wind components would be larger than vertical wind components in slant pointing Doppler spectra measurements, it would be good if components of horizontal wind could be removed from each Doppler spectrum plot, so that readers can track growth processes which can be represented by increases/decreases of reflectivity and abso-

lute values of Doppler velocity.

Specific comments:

1) I was confused by increase/decrease of fall speed when looking at the observed spectra changing with height. The TARA-observed Doppler spectra include horizontal wind component in addition to particle fall speed component. From the observed spectra (e.g., Figures 7, 11, and 13), it was difficult to see particle growth, which can be represented by particle fall speeds, because the spectra included large components of horizontal wind. I recommend extracting the horizontal wind component from the observed spectra. I think that this is not so difficult because the authors nicely retrieved horizontal wind.

2) Polarimetric variables have an elevation dependency; for instance, Zdr values decrease with elevation angle for horizontally-oriented oblate particles. Particularly, the Zdr values can significantly decrease above an elevation angles of 20 degrees. Did you correct the observed Zdr for elevation angles?

3) Section 4, Figure 3: As the authors mentioned, the differential reflectivity is influenced by particle densities. One example is that aggregation can reduce its density, resulting in decreasing in Zdr (this was mentioned in the text). Another example is that initial riming of branched crystals can increase the density as gaps between branches are filled, resulting in increase in Zdr. I recommend mentioning this effect in the text as well.

4) Section 5: Please explain how to take into account individual particle fall speeds to retrieve fall streaks and discuss particle growth of individual particle populations. Particles included in the radar sampling volume have different fall speeds. In the next range bin, the composition of particles in the volume can be different from that in the previous range bin volume above, because individual particles can have different fall speeds (i.e. size sorting effect). This is true even for retrieved fall streaks. When discussing ice particle growth using Doppler spectra at different heights (Figures 7, 11,

and 13), I think that different particle fall speeds should be considered. Please explain if some assumptions were used in the discussion.

5) P. 8, line 22: There could be non-Rayleigh scattering effect in addition to attenuation.

6) P. 8, line 31 "homogeneous wind": Does this mean horizontally homogeneous?

7) P. 8, line 32 "shear": vertical shear?

8) P. 10, lines3-4 "The closer...": If large particles dominated the total reflectivity, RHOhv may not reflect the particle diversity. In that case, as overall there is little contribution from the non-spherical particles, resulting in high RHOhv.

9) Section 6.1, Figures 6 and 10: What is the minimum limitation value of LDR due to the antenna limitation? In Figures 6 and 10, below Region N, LDR seems to be relatively high (∼-25 dB) at the edges of spectra. LDR tends to be large with low signal-to-noise ratio. What can the relatively high LDR at the edges of spectra indicate?

10) Section 6.1:, Figure 7: Compared to other studies showing S-band polarimetric radar Zdr in dendritic growth zones (e.g., Kumjian and Lombardo, 2017, doi: 10.1175/MWR-D-15-0451.1; Griffin et al. 2018, doi: 0.1175/JAMC-D-17-0033.1), Zdr values in Fig. 7 are relatively small. Why? Is there an elevation dependency?

11) Figure 9: How did the radiosondes measure supercooled liquid droplets? Did they have special sensors?

12) P. 10, line 19: What is the difference between ice particles and snowflakes here? I guess this meant ice crystals and snowflakes (aggregates)?

13) P. 11, lines 6-9: This does not make sense to me. I am wondering why the seeded case showed slower increase in Zh? I think that the ice seeding could accelerate aggregation, resulting in rapid increase in Zh...

14) P. 11, line 17: To me, the spectrum at 3.1 km does not seem to broaden (Fig. 10c). Could you snow a zoomed up plot?

15) P. 11, line 34: Toward 2864 m in Figure 11, sZ values increases, while sZdr kept their values. Does this profile suggest increase in number concentration rather than size? What is the source of nucleated ice?

16) P. 12, line 1: Please mention effects of horizontal wind components. Do the particle populations having Doppler velocity >-8.5 at 3055 m exactly correspond to those at 2864 m?

17) P. 12, line 8, Figure 11: Significant negative values in Zdr were also shown at 3055 m. Could you explain the negative values at this altitude?

18) Section 6.2, P. 12, line 22: I am not sure why the authors identified the Zdr signature as needles/columns and why they decided that the TARA-observed Zdr corresponded to the Mira-observed Ldr. As the authors pointed out, the retrieved Zdr profile and the t_0 profile were inconsistent at the region N in Figure 9. This suggested that the TARA radar measurements and Mira radar measurements looked at different locations and different particles.

---

## Referee Comment (RC2) · Anonymous Referee #1 · 12 Mar 2018

This paper reports spectral polarisation radar measurements within a rather complex ice cloud system, and subjects these data to careful analysis using a "fall streak reconstruction" technique. The main virtue of the paper is the effort to pull together all the measurements to reveal a story about the evolution of the ice particles. This story is qualitative rather than quantitative, but this is fair enough - most other studies of this kind in the literature have the same limitation.

I think this is well within ACP's scope. It's pretty well-written, the results and methodology used are of interest to the community, and the paper makes a worthwhile addition to the existing literature on this topic. I have a number of minor corrections below - I

think the paper is publishable once these have been addressed.

Introduction

page 2, line 8 onwards. You make a big deal here about how it's important to study precipitating mixed-phase clouds and melting of ice particles, linking the ice above the melting layer to the rain below. I saw very few links in your paper to this aspect. Essentially your focus is on the ice-phase bit of a cloud that happens to be raining at the surface. I suggest that you change the emphasis a bit here to fit better with what you are doing.

You also make a link here to attenuation and attempt to establish a dichotomy between cloud radars (W, Ka, X bands) and precipitation radars (C- and S-band), then latter being relatively immune to attenuation. This seems to be a link to motivate the use of TARA which is S-band. I personally think the distinction is overplayed, but I acknowledge that the longer wavelength has some advantages for the interpretation of the data. I'm not sure I would make such a big deal of it myself, but I leave that to the authors.

Studies by Bader et al, Field et al were S-band not C-band.

You might consider adding Keat et al (2017, JGR) to your review of relevant literature here. http://onlinelibrary.wiley.com/doi/10.1002/2017JD026754/abstract again identified oriented pristine crystals and inferred the presence of supercooled liquid water . Seems relevant to your case 2.

Section 2.

page 4, line 30,31. You say cloud top from MIRA is at least 0.5km higher than the one from TARA. Always? Surely this is case-dependent. I suggest it is better to quote the minimum detectable dBZ at, say, 5km height for the two radars.

Section 3.

The arrow at the bottom of figure 2 confused me. First because it points backwards compared to the evolution of the particles in the cartoons. Second because it implies ZDR goes down as the particles grow, but this is not the case in your vapour deposition cartoon where ZDR is maintained. So I suggest removing this.

Orientation of particles. This issue first arises in section 3 (line 14, page 5) but continues throughout the paper. The terminology you use here mixes two distinct physical characteristics: shape (where the idea of prolate, oblate makes some kind of sense) and orientation (which is completely distinct and controlled by aerodynamics). You use the term "prolate" to mean a particle which is broadly in the shape of a prolate spheroid, which has its long axis vertical. Conical graupel can be an example of this. But there are other particles (like needles) which are prolate in shape but have a horizontal preferred orientation. So I strongly recommend you disentangle shape and orientation here and in the rest of the paper.

Page 5, line 18. Say diffusional growth of ice happens when vapour diffuses towards the crystals instead of forming supercooled droplets. Why instead? It is perfectly possible to grow both if the supersaturation is large enough.

line 20 - during diffusional growth particles keep their characteristic shape. I would like some clarification on this. I can think of two obvious counter examples: (1) dendrites for example often grow wider and wider without thickening significantly, and can end up with aspect ratios of 100:1. (2) when an in particle falls into a different temperature from its earlier growth - e.g. rosettes grow plate-like appendages, columns get plate caps, etc. So perhaps you can be more direct in your meaning here, and what you are assuming which is actually critical for the analysis that follows vs general information

line 23-30 These are relevant bits of literature, but personally I think the evidence is not so clear cut. Suggest softening the wording in this paragraph to explain these studies have suggested or indicated what nature might be doing. Then when you pick these ideas up later and find evidence to support rapid aggregation of needles for example,

that also makes the contribution of your new results more apparent.

Section 4.

Page 6 line 2. The reference to the text books here is not necessary, you have already introduced these processes. There are a few other places in the text where I felt the referencing was repetitive (e.g. Westbrook et al 2007 on page 10, line 20)

line 17-19 - recommend remove discussion about sZDR in rain. It is irrelevant to what follows and confused me when I was trying to understand this paragraph.

line 23-29. I found this hard to understand. Is diffusion dominant in this scenario? or aggregation? or both? could be clearer. the first line of the paragraph says "a signature of diffusional growth", but then there is lots of talk of aggregation.

Figure 3 could potentially be annotated to help make it clearer - e.g. with text and arrows saying for example things like "rimed particles" "crystals growing from vapour" etc

I think generally in section 4 it would help to be clear that this is your conceptual picture of how these processes play out in a typical cloud, rather than asserting these signatures are universal - which would require detailed evidence

Section 5.

Fall streak reconstruction. Any strengths and limitations to the technique worth summarising here? (for reader who does not go back to your previous paper)

Page 8 line 13. "3 time bins" - can you specify in seconds please

Section 6.

Page 8, line 18. "S-band radar profiler TARA" - repetitive. Already introduced.

line 31-32. homogeneous wind conditions. can you be quantitative?

Panels in figure 6, 10, 12 could be neater. "differential reflectivity [dB]" label on fig 6

panel (e) is impinging on the panel next to it (f). Quite a mix of font sizes etc.

Page 10, line 10. "a supercooled liquid water layer... is identified". How? From RH close to 100%? or some more sophisticated balloon-borne sensor. If the former do need to acknowledge this is not direct (but I do believe it)

Talk a lot about needles in case 1. How do you define a needle? Do you count a hexagonal column as a needle?

In case 2 and case 3 there are parts of the sZDR spectra which are negative. However you also acknowledge that there is a lot of fluctuation in these sZDR values. This raises the question - what is the expected random fluctuation on a sZDR data point? Can you estimate that? I think that would help the discussion a lot if you could.

Page 12, line 27. "newly generated particles... lead to an increase of Z in the rain pattern below the melting layer". This wasn't obvious to me. Can you be more explicit how you determined this to be the case?

Page 13, line 13. 114730UTC. Similar format for time elsewhere. I've not met this way of expressing time before. I guess it is just HHMMSS but you mix this around in the text with the more usual HHMM. Can I suggest you clarify this somewhere.

Page 13, line 4 (and elsewhere). When you do the fall streak reconstruction, you talk about using a "cloud base height" of 2.25km (for example). These aren't really cloud base heights though (CBH is a physical characteristic of the cloud) - instead they are simply boundary conditions for the reconstruction. So suggest rephrase. In case 3 the choice of 2.25km seems almost immediately invalidated by your arguments for not analysing any data below 3km. Can you justify this?

line 16. Aggregation. So aggregation is occurring 5-4km. But then why does it stop below that?

---

## Author Comment (AC1) · 26 Mar 2018

This paper reports spectral polarisation radar measurements within a rather complex ice cloud system, and subjects these data to careful analysis using a "fall streak reconstruction" technique. The main virtue of the paper is the effort to pull together all the measurements to reveal a story about the evolution of the ice particles. This story is qualitative rather than quantitative, but this is fair enough - most other studies of this kind in the literature have the same limitation.

I think this is well within ACP's scope. It's pretty well-written, the results and methodology used are of interest to the community, and the paper makes a worthwhile addition

to the existing literature on this topic. I have a number of minor corrections below - I think the paper is publishable once these have been addressed.

We would like to thank the reviewer for the time and effort provided for the review. Lukas Pfitzenmaier, Christine Unal, Yann Dufournet and Herman Russchenberg.

Introduction page 2, line 8 onwards. You make a big deal here about how it's important to study precipitating mixed-phase clouds and melting of ice particles, linking the ice above the melting layer to the rain below. I saw very few links in your paper to this aspect. Essentially your focus is on the ice-phase bit of a cloud that happens to be raining at the surface. I suggest that you change the emphasis a bit here to fit better with what you are doing.

Answer: The focus is indeed given to the precipitating mixed-phase cloud. However in the investigated cases, the rain below is constantly taken into account as input to rearrange the reflectivity data based on the fall streak method (bottom-up approach, Section 5).

You also make a link here to attenuation and attempt to establish a dichotomy between cloud radars (W, Ka, X bands) and precipitation radars (C- and S-band), then latter being relatively immune to attenuation. This seems to be a link to motivate the use of TARA which is S-band. I personally think the distinction is overplayed, but I acknowledge that the longer wavelength has some advantages for the interpretation of the data. I'm not sure I would make such a big deal of it myself, but I leave that to the authors.

Answer: Thank you for this opinion. As mentioned above, the emphasis of this study is to analyze convective/stratiform rainfall cases with potentially high reflectivity values, and severe reflectivity difference due to attenuation (see comparison MIRA and TARA in Figure 4).

Studies by Bader et al, Field et al were S-band not C-band.

Answer: This is corrected in the paper.

You might consider adding Keat et al (2017, JGR) to your review of relevant literature here. http://onlinelibrary.wiley.com/doi/10.1002/2017JD026754/abstract again identified oriented pristine crystals and inferred the presence of supercooled liquid water. Seems relevant to your case 2.

Answer: Thank you for mentioning Keat and Westbrook, 2017, we missed this article, which is now read and referenced in the article.

Section 2.

page 4, line 30,31. You say cloud top from MIRA is at least 0.5km higher than the one from TARA. Always? Surely this is case-dependent. I suggest it is better to quote the minimum detectable dBZ at, say, 5km height for the two radars.

Answer: We emphasized in the text the case dependency of this observation.

Section 3.

The arrow at the bottom of figure 2 confused me. First because it points backwards compared to the evolution of the particles in the cartoons. Second because it implies ZDR goes down as the particles grow, but this is not the case in your vapour deposition cartoon where ZDR is maintained. So I suggest removing this.

Answer: Acknowledged.

Orientation of particles. This issue first arises in section 3 (line 14, page 5) but continues throughout the paper. The terminology you use here mixes two distinct physical characteristics: shape (where the idea of prolate, oblate makes some kind of sense) and orientation (which is completely distinct and controlled by aerodynamics). You use the term "prolate" to mean a particle which is broadly in the shape of a prolate spheroid, which has its long axis vertical. Conical graupel can be an example of this. But there are other particles (like needles) which are prolate in shape but have a horizontal preferred orientation. So I strongly recommend you disentangle shape and orientation here and in the rest of the paper.

Answer: Thank you for this comment. The text is adapted to be more specific on this topic.

Page 5, line 18. Say diffusional growth of ice happens when vapour diffuses towards the crystals instead of forming supercooled droplets. Why instead? It is perfectly possible to grow both if the supersaturation is large enough.

Answer: The text is adapted.

line 20 - during diffusional growth particles keep their characteristic shape. I would like some clarification on this. I can think of two obvious counter examples: (1) dendrites for example often grow wider and wider without thickening significantly, and can end up with aspect ratios of 100:1. (2) when an in particle falls into a different temperature from its earlier growth - e.g. rosettes grow plate-like appendages, columns get plate caps, etc. So perhaps you can be more direct in your meaning here, and what you are assuming which is actually critical for the analysis that follows vs general information.

Answer: The text is adapted.

line 23-30 These are relevant bits of literature, but personally I think the evidence is not so clear cut. Suggest softening the wording in this paragraph to explain these studies have suggested or indicated what nature might be doing. Then when you pick these ideas up later and find evidence to support rapid aggregation of needles for example, that also makes the contribution of your new results more apparent.

Answer: The text is adapted.

Section 4.

Page 6 line 2. The reference to the text books here is not necessary, you have already introduced these processes. There are a few other places in the text where I felt the

referencing was repetitive (e.g. Westbrook et al 2007 on page 10, line 20) line 17-19 - recommend remove discussion about sZDR in rain. It is irrelevant to what follows and confused me when I was trying to understand this paragraph.

Answer: The references are removed as well as the discussion of the sZdr in rain.

line 23-29. I found this hard to understand. Is diffusion dominant in this scenario? Or aggregation? or both? could be clearer. the first line of the paragraph says "a signature of diffusional growth", but then there is lots of talk of aggregation.

Answer: The first line is now: Figure 3 b) depicts a signature of diffusional growth and aggregation in sZ and sZdr.

Figure 3 could potentially be annotated to help make it clearer - e.g. with text and arrows saying for example things like "rimed particles" "crystals growing from vapour" etc

Answer: We finally kept the Figure as it was, because the additional text made the Figure too busy and therefore unclear.

I think generally in section 4 it would help to be clear that this is your conceptual picture of how these processes play out in a typical cloud, rather than asserting these signatures are universal - which would require detailed evidence.

Answer: The last two sentences of the first paragraph of section 4 is now: It is pointed out that these sketches are meant to explain spectral signatures of a S-band slantwise profiling radar. Other radar setups may have different spectral signatures.

Section 5.

Fall streak reconstruction. Any strengths and limitations to the technique worth summarising here? (for reader who does not go back to your previous paper)

Answer: A small section is added at the end of the first paragraph of page 8.

Page 8 line 13. "3 time bins" - can you specify in seconds please

Answer: This is done.

Section 6.

Page 8, line 18. "S-band radar profiler TARA" - repetitive. Already introduced.

Answer: The repetition is removed.

line 31-32. homogeneous wind conditions. can you be quantitative?

Answer: The values of the retrieved horizontal wind are in average around 22 m/s in the cloud. However, one can see that in some regions values up to 25 m/s are visible. Due to the fact that values up to 35 m/s are visible when the new air mass is moving upwards and the wind direction changes (Case 3), we assume the horizontal wind field in this case as homogeneous only above 3 km.

Panels in figure 6, 10, 12 could be neater. "differential reflectivity [dB]" label on fig 6 panel (e) is impinging on the panel next to it (f). Quite a mix of font sizes etc.

Answer: Differential reflectivity [dB] label on fig 6 panel (e) doesn't impinge anymore on the panel next to it (f).

Page 10, line 10. "a supercooled liquid water layer...is identified". How? From RH close to 100%? or some more sophisticated balloon-borne sensor. If the former do need to acknowledge this is not direct (but I do believe it).

Answer: Super-cooled liquid water is assumed when the temperature and the dew point temperature of the radiosonde launch match. In general, this is an indication for the supersaturation of water vapor in these areas. In the ice phase of clouds supersaturation of liquid water is also possible, which is assumed here if the relative humidity reaches 100%. This is shown in the light blue shaded areas in the Figure 9.

Talk a lot about needles in case 1. How do you define a needle? Do you count a

hexagonal column as a needle?

Answer: A needle is defined as a prolate ice particle. An explicit distinction between needles and hexagonal columns using polarimetric radar observations is rather complicated and almost impossible. They both originate in regions with similar temperatures. Needles are generated if super cooled liquid is present while hexagonal columns are formed under supersaturation with respect to ice. The identification of needles is based on the given temperature and supersaturation detected by the radiosonde. The MIRA measurements confirm such particle shapes. However a 100% prove can only be given using in situ observations in this case. Nonetheless we are quite sure that our synergistic observation gives a good indication that we observed needle during Case 1.

In case 2 and case 3 there are parts of the sZDR spectra which are negative. However you also acknowledge that there is a lot of fluctuation in these sZDR values. This raises the question - what is the expected random fluctuation on a sZDR data point? Can you estimate that? I think that would help the discussion a lot if you could.

Answer: The variance of sZdr is large, and increase when the SNR and the copolar correlation coefficient decrease. To mitigate this issue, only data with SNR larger than 10 dB are considered. Further 3 consecutive Doppler spectra are averaged in time (hh and vv) for this study to obtain consistent trends in sZdr at different times and heights. Significant negative values are obtained in this case. It might be that during the growth process and due to turbulence some prolate particles are within the volume (sZdr spectrum at 2864 m). Nevertheless, the large negative sZdr are rather uncommon and there is a sharp decrease of the sZdr values at the edge of the spectrum, making them questionable. For the presented spectrum at 3055 m, due to nucleation and growth of the seeded particle from above the probability of prolate particle in the volume is higher. Also the drop into negative values is less sharp.

Page 12, line 27. "newly generated particles...lead to an increase of Z in the rain

pattern below the melting layer". This wasn't obvious to me. Can you be more explicit how you determined this to be the case?

Answer: By analyzing the time height plots (Fig 4) and the spectral data discussed in Section 6 an increase of reflectivity in rain was observed. This increase seems to have a strong correlation with the Zdr band above that can be related to newly formed pristine ice particles (with defined shapes). One can also see a decrease on rain reflectivity when the discussed signatures (Zdr band) are not observed.

Page 13, line 13. 114730UTC. Similar format for time elsewhere. I've not met this way of expressing time before. I guess it is just HHMMSS but you mix this around in the text with the more usual HHMM. Can I suggest you clarify this somewhere.

Answer: The time information will be unified and clarified in the text.

Page 13, line 4 (and elsewhere). When you do the fall streak reconstruction, you talk about using a "cloud base height" of 2.25km (for example). These aren't really cloud base heights though (CBH is a physical characteristic of the cloud) - instead they are simply boundary conditions for the reconstruction. So suggest rephrase. In case 3 the choice of 2.25km seems almost immediately invalidated by your arguments for not analysing any data below 3km. Can you justify this?

Answer: The cloud base height is selected above the melting layer in precipitating cloud systems. The fall streak retrieval is optimized to analysis the cloud containing ice particles. Concerning Case 3, the data below 3 km are not analyzed because of the horizontal wind direction shear. In that case we cannot relate the microphysical properties of the particle population under investigation to the ones below 3 km related probably to another particle population. However the full fall streak retrieval can be obtained (homogeneous conditions are not required), but in that case cannot be fully exploited.

line 16. Aggregation. So aggregation is occurring 5-4km. But then why does it stop

below that?

Answer: From the observations the main growth of the particles occurs between 5 km and 4 km. Considering the MIRA Doppler power spectra, the mean Doppler velocity slowly increases towards the melting layer, which means that aggregation continues (slowly) until the melting layer, as we could expect.

Please also note the supplement to this comment:
https://www.atmos-chem-phys-discuss.net/acp-2017-1032/acp-2017-1032-AC1-supplement.pdf

[Figure]

**Supplement:**

[revised manuscript text omitted]

---

## Author Comment (AC2) · 26 Mar 2018

General Comment: This study collected polarimetric Doppler spectra at an elevation angle of 45 degrees. The observed spectra were realigned with height along with retrieved fall streaks and analyzed the reflectivity and differential reflectivity spectra changing with height to discuss ice particle growth. The novelty technique and idea used in this study are very interesting, but sometimes I was confused by increase/decrease of fall speed when looking at the observed spectra. Because horizontal wind components would be larger than vertical wind components in slant pointing Doppler spectra measurements, it would be good if components of horizontal wind

could be removed from each Doppler spectrum plot, so that readers can track growth processes which can be represented by increases/decreases of reflectivity and absolute values of Doppler velocity.

We would like to thank the reviewer for the time and effort provided for the review. Lukas Pfitzenmaier, Christine Unal, Yann Dufournet, and Herman Russchenberg

1) I was confused by increase/decrease of fall speed when looking at the observed spectra changing with height. The TARA-observed Doppler spectra include horizontal wind component in addition to particle fall speed component. From the observed spectra (e.g., Figures 7, 11, and 13), it was difficult to see particle growth, which can be represented by particle fall speeds, because the spectra included large components of horizontal wind. I recommend extracting the horizontal wind component from the observed spectra. I think that this is not so difficult because the authors nicely retrieved horizontal wind.

Answer 1) We agree than the removal of the horizontal wind would help in the interpretation of the measured Doppler spectra. The implementation in the original manuscript was considered during the writing process. Because the results were not as expected we decided to show the not corrected spectra. One reason for this is the measurement geometry of the TARA radar and the design of wind retrieval. In the wind retrieval at high time resolution, homogeneous conditions with the 3 probing beams is assumed, which is not the case for all cloud conditions. This leads to some problems especially for dynamically inhomogeneous cloud systems as discussed in the paper. Second reason is, that at that stage, we did not remove the contribution of the mean horizontal wind in the measured Doppler velocities. If we would correct the Doppler velocities for the mean horizontal wind, we have still in the Doppler velocity measurement, a residual component of the horizontal wind (difference between the actual horizontal wind and the mean horizontal wind) AND the actual vertical wind AND the actual Doppler fall velocity. We show non-averaged spectra. Therefore, presently, we cannot provide the Doppler fall velocity, which would be, of course very useful to interpret growth

processes.

2) Polarimetric variables have an elevation dependency; for instance, Zdr values decrease with elevation angle for horizontally-oriented oblate particles. Particularly, the Zdr values can significantly decrease above an elevation angles of 20 degrees. Did you correct the observed Zdr for elevation angles?

Answer 2) The shown Zdr values are not corrected for elevation at which they are observed. Also, we perform only qualitative analyzes of the values and relate the values to a specific particle type. Therefore, we did not correct the values for elevation.

3) Section 4, Figure 3: As the authors mentioned, the differential reflectivity is influenced by particle densities. One example is that aggregation can reduce its density, resulting in decreasing in Zdr (this was mentioned in the text). Another example is that initial riming of branched crystals can increase the density as gaps between branches are filled, resulting in increase in Zdr. I recommend mentioning this effect in the text as well.

Answer 3) Indeed riming of branched crystals increases the particle density. This leads to an increase of the Zdr-values. This scenario is not implemented in the article. We focus on the possible scenarios of growth processes which can explain the discussed measurements.

4) Section 5: Please explain how to take into account individual particle fall speeds to retrieve fall streaks and discuss particle growth of individual particle populations. Particles included in the radar sampling volume have different fall speeds. In the next range bin, the composition of particles in the volume can be different from that in the previous range bin volume above, because individual particles can have different fall speeds (i.e. size sorting effect). This is true even for retrieved fall streaks. When discussing ice particle growth using Doppler spectra at different heights (Figures 7, 11, and 13), I think that different particle fall speeds should be considered. Please explain if some assumptions were used in the discussion.

Answer 4) This effect is not taken into account in the fall streak retrieval, see Pfitzen-maier et al., 2017, doi:10.1175/JTECH-D-16-0117.1. The fall streak retrieval is based on Doppler measurements (mean Doppler velocities) and do not take into account the distribution of Doppler velocities. Therefore the retrieval consists of a mean fall streak. In the fall streak retrieval article, Section 4a explains the limitations of the horizontal wind retrieved by TARA, which have an impact on the fall streak retrieval. With the current retrieval the mean movement of the particle population can be tracked. Therefore size sorting cannot be taken into account, if it occurs.

5) P. 8, line 22: There could be non-Rayleigh scattering effect in addition to attenuation.

Answer 5) The contribution to non-Rayleigh scattering effects in addition to the attenuation is added to the text.

6) P. 8, line 31 "homogeneous wind": Does this mean horizontally homogeneous?

Answer 6) Yes, horizontally homogeneous wind conditions are meant. This will be clarified in the text

7) P. 8, line 32 "shear": vertical shear?

Answer 7) Yes, this is a vertical shear in the wind direction (about 30 deg.). This will be clarified in the text

8) P. 10, lines3-4 "The closer...": If large particles dominated the total reflectivity, RHOhv may not reflect the particle diversity. In that case, as overall there is little contribution from the non-spherical particles, resulting in high RHOhv.

Answer 8) The statement of the reviewer is right. However, in this case, considering the spectral polarimetric signature, sZdr, which is flat versus Doppler velocity (Fig. 7, from 3076 m to 2524 m), the particles are spherical independently of their size. Therefore, the high RHOhv values are observed in the area 3076-2524 m. The text will be adjusted.

[Figure]

9) Section 6.1, Figures 6 and 10: What is the minimum limitation value of LDR due to the antenna limitation? In Figures 6 and 10, below Region N, LDR seems to be relatively high (âĹij -25 dB) at the edges of spectra. LDR tends to be large with low signal-to-noise ratio. What can the relatively high LDR at the edges of spectra indicate?

Answer 9) First we want to point out that a 10 dB SNR clipping was applied to the Mira sLdr to avoid contributions from low SNR regions near the edges of the spectra. The technical limitation for the Ldr detection is at -35 dB. Tyynelä et al, 2011, (doi.org/10.1175/JTECH-D-11-00004.1) modeled Ldr for vertical pointing radar for a range of frequencies. There it was found that Ldr in the simulations vary more than expected for the two different model approaches discussed. It was found that aggregates seem to produce larger Ldr than smaller ice crystals. One reason they pointed out may be incorrect mass size relation for aggregates. Nevertheless, aggregates have complex shapes and align during the sedimentation in a preferred orientation. Analyzing the Mira spectra, we observed large aggregates (Doppler velocity up to 2 m/s) which can explain the increased Ldr for larger particles. While smaller particles have still a more defined shape and therefore increased Ldr values (modeled values cited in the paper).

10) Section 6.1:, Figure 7: Compared to other studies showing S-band polarimetric radar Zdr in dendritic growth zones (e.g., Kumjian and Lombardo, 2017, doi: 10.1175/MWR-D-15-0451.1; Griffin et al. 2018, doi: 0.1175/JAMC-D-17-0033.1), Zdr values in Fig. 7 are relatively small. Why? Is there an elevation dependency?

Answer 10) Zdr values are indeed relatively small because the measurement is carried out at 45 deg. elevation. Again a trade-off between polarimetric signature and Doppler velocities related to Doppler fall velocities. Although we don't have yet the absolute values of terminal fall velocities.

11) Figure 9: How did the radiosondes measure supercooled liquid droplets? Did they have special sensors?

Answer 11) The radiosondes are standard Vaisala sondes and did not have special sensors. Super-cooled liquid water is assumed when the temperature and the dew point temperature of the radiosonde launch match. In general, this is an indication for the supersaturation of water vapor in this area. In the ice phase of clouds supersaturation of liquid water is also possible, which is assumed here if the relative humidity reaches 100%. This is shown in the light blue shaded areas in the Figure 9.

12) P. 10, line 19: What is the difference between ice particles and snowflakes here? I guess this meant ice crystals and snowflakes (aggregates)?

Answer 12) It means that pristine ice crystals or small aggregates grow via aggregation into larger and denser aggregates and snowflakes, respectively.

13) P. 11, lines 6-9: This does not make sense to me. I am wondering why the seeded case showed slower increase in Zh? I think that the ice seeding could accelerate aggregation, resulting in rapid increase in Zh...

Answer 13) It is right that the increase of the Zh slope is less in Case 2 than in Case 1. Also other observations suggest that the aggregation efficiency is less than in Case 1. The increase of the observed mean Doppler velocity is less than during Case 1, in the Mira measurements. This leads to the assumption that falling ice particles have lower density and might be smaller. Reason for that can be lower supersaturation in that height and lower concentration of generated particles at around 3100 m. In the cited paper by Hobbs et al., 1974, a strong relation between the particle number concentration and aggregation efficiency for needles is mentioned. Lower number concentrations would lead to less dense aggregates. Also we do not know the aggregation efficiencies of the seeding and the generated particles. From the observations we cannot confirm the fact that the ice seeding could accelerate aggregation, resulting in rapid increase in Zh. However, to investigate this further in situ measurements or additional sensors would have been needed to compare these observations.

14) P. 11, line 17: To me, the spectrum at 3.1 km does not seem to broaden (Fig. 10c).

[Figure]

Could you snow a zoomed up plot?

Answer 14) Fig. 11 shows the single spectra corresponding to Fig. 10c and e. There it can be seen that the spectra broadens from 3394 m to 3055 m from ~1 m/s up to 1.5 m/s. This in not clearly visible in the spectrograms in Fig. 10. Therefore, the single spectra are shown to give a better and more detailed view into the growth process region, while the spectrograms represent the whole fall streak.

15) P. 11, line 34: Toward 2864 m in Figure 11, sZ values increases, while sZdr kept their values. Does this profile suggest increase in number concentration rather than size? What is the source of nucleated ice?

Answer 15) It is right that the single sZ bins increase more than the spectrum broadens. Therefore, it is right that one could assume that the particle number concentration in that region increases more that the size. This could be due to an ongoing ice multiplication process or to a continuous and ongoing particle generation process. The small ice crystals would grow in size by diffusional growth and keep their size dependence. However, the increase of the single sZ-bins is too large to be explained by only an increase of number concentration. As already pointed out, TARA spectra are not corrected for the mean horizontal wind contribution and therefore no direct link between Doppler velocity and particle size can be drawn. Nevertheless, the vertical pointing Mira shows such an increase of Doppler velocities towards the melting layer and we assumed such an increase for the TARA spectra as well. Assuming a diffusional growth of the smaller particles before aggregation leads to less dense and slightly smaller particles, which may cause such spectral signatures. As also already mentioned, because of the lack of additional information we cannot give more insights into the discussed case.

16) P. 12, line 1: Please mention effects of horizontal wind components. Do the particle populations having Doppler velocity >-8.5 at 3055 m exactly correspond to those at 2864 m?

Answer 16) It is clearly stated in the captions of Figures 6, 10 and 12 that the Doppler

velocity contains the radial wind. Therefore we have to interpret the Doppler velocity only relatively. Concerning Case 2 (altitudes 3055 m and 2864 m), we cannot neglect dynamical influences, although the wind direction at these heights is almost constant.

17) P. 12, line 8, Figure 11: Significant negative values in Zdr were also shown at 3055m. Could you explain the negative values at this altitude?

Answer 17) The variance of sZdr is large, and increase when the SNR and the copolar correlation coefficient decrease. To mitigate this issue, only data with SNR larger than 10 dB are considered. TARA processing provides Doppler spectra with the average number 2. Further 3 of these Doppler spectra are averaged (hh and vv) for this study to obtain consistent trends in sZdr at different times and heights. The total number of averaging is thus 6. Significant negative values are obtained in this case. It might be that during the growth process and due to turbulence some prolate particles are within the volume (sZdr spectrum at 2864 m). Nevertheless, the large negative sZdr are rather uncommon and there is a sharp decrease of the sZdr values at the edge of the spectrum, making them questionable. For the presented spectrum at 3055 m, due to nucleation and growth of the seeded particle from above the probability of prolate particle in the volume is higher. Also the drop into negative values is less sharp.

18) Section 6.2, P. 12, line 22: I am not sure why the authors identified the Zdr signature as needles/columns and why they decided that the TARA-observed Zdr corresponded to the Mira-observed Ldr. As the authors pointed out, the retrieved Zdr profile and the $t\_0$ profile were inconsistent at the region N in Figure 9. This suggested that the TARA radar measurements and Mira radar measurements looked at different locations and different particles.

Answer 18) The hypothesis of the presence of needles/columns is mainly built upon the radiosonde temperature range at the considered altitudes. The radar MIRA is used to confirm this hypothesis, although we know that both radars measure different sampling volumes.

Please also note the supplement to this comment:
https://www.atmos-chem-phys-discuss.net/acp-2017-1032/acp-2017-1032-AC2-supplement.pdf

―――――――――――――――――――

[Figure]

**Supplement:**

[revised manuscript text omitted]